# Bayesian Optimization of Antibodies Informed by a Generative Model of Evolving Sequences

**Alan Nawzad Amin**
New York University

**Nate Gruver***
New York University

**Yilun Kuang***
New York University

**Lily Li***
New York University

**Hunter Elliott**
BigHat Biosciences

**Calvin McCarter**
BigHat Biosciences

**Aniruddh Raghu**
BigHat Biosciences

**Peyton Greenside**
BigHat Biosciences

**Andrew Gordon Wilson**
New York University

## Abstract

To build effective therapeutics, biologists iteratively mutate antibody sequences to improve binding and stability. Proposed mutations can be informed by previous measurements or by learning from large antibody databases to predict only typical antibodies. Unfortunately, the space of typical antibodies is enormous to search, and experiments often fail to find suitable antibodies on a budget. We introduce Clone-informed Bayesian Optimization (CloneBO), a Bayesian optimization procedure that efficiently optimizes antibodies in the lab by teaching a generative model how our immune system optimizes antibodies. Our immune system makes antibodies by iteratively evolving specific portions of their sequences to bind their target strongly and stably, resulting in a set of related, evolving sequences known as a *clonal family*. We train a large language model, CloneLM, on hundreds of thousands of clonal families and use it to design sequences with mutations that are most likely to optimize an antibody within the human immune system. We propose to guide our designs to fit previous measurements with a twisted sequential Monte Carlo procedure. We show that CloneBO optimizes antibodies substantially more efficiently than previous methods in realistic *in silico* experiments and designs stronger and more stable binders in *in vitro* wet lab experiments.

## 1 Introduction

Antibody therapeutics are the fastest growing class of drugs, with approved treatments for a breadth of disorders ranging from cancer to autoimmune disease to infectious disease (Carter & Lazar, 2018). Biologists wish to design antibodies that strongly bind to targets of interest while being stable in the human body. Stable antibodies do not unfold or cause an adverse immune reaction (Jarasch et al., 2015). To develop these antibodies, biologists first screen many diverse antibody sequences, or use a lab animal's immune system to find an initial candidate that binds a target. This candidate often does not bind strongly or is unstable in the human body, so it is used as a starting point in an iterative optimization experiment in which biologists predict mutations that result in better or more stable binders (Lu et al., 2020).

To make these predictions, we can learn from up to thousands of sequence measurements from many previous iterations (Rapp et al., 2024; Yang et al., 2019; Fannjiang et al., 2022; Brookes et al., 2019). We can also learn from databases of protein sequences to avoid predicting mutations that produce nonfunctional antibodies (Gruver et al., 2023; Stanton et al., 2022; Hie et al., 2023; Prihoda et al., 2022). However, even with this restriction, there are a combinatorial number of mutations we could predict, only a handful of which are beneficial. Therefore, optimization experiments regularly fail to find suitable sequences on a budget.

To optimize more efficiently than current methods, we need an informed prior about where and how to mutate to positively affect binding and stability. Ideally we could learn what mutations often lead to better sequences in optimization experiments in the lab. Unfortunately such data is scarce. In principle, we can instead learn from abundant data about what mutations often lead to better

---

*Equal contribution

Figure 1: Our immune system introduces mutations (blue) to evolve weak binders of a target into strong binders (green). The result is a set of related sequences that bind the antigen strongly and stably known as a clonal family. We use a model trained on these families, CloneLM, to perform Bayesian optimization in a procedure called CloneBO. We use experimental data to generate a clonal family that might have evolved to bind our antigen and suggest sequences to test in the lab.

sequences in our bodies. To make an antibody that binds a new target, our immune system evolves sets of related sequences known as *clonal families*; through selection, sequences in clonal families accumulate mutations that increase binding to a target while maintaining stability (Burnett et al., 2018). Through large-scale sequencing efforts, we can now learn from databases that contain large numbers of these evolving sequences (Olsen et al., 2022a).

In this paper, we introduce Clone-informed Bayesian optimization (CloneBO), a Bayesian optimization procedure which efficiently optimizes antibody sequences in the lab by teaching a generative model how the human immune system optimizes antibodies (Fig. 1). In Section 2 we review related work. In Section 3 we introduce the problem of iterative Bayesian optimization. In Section 4 we describe how in theory we can build a prior for where and how to mutate given observed clonal families. In Section 5 we build such a prior in practice by fitting a large language model, CloneLM, to hundreds of thousands of clonal families. We take a martingale posterior approach to sampling in which we generate new clonal families that contain our candidate. In Section 6 we describe how to condition on previous measurements using a twisted sequential Monte Carlo procedure so that good mutations are included in our clonal family, and bad mutations are excluded. We use our model to build a Bayesian optimization procedure, CloneBO. In Section 7 we show that CloneBO optimizes realistic oracles for stability and binding strength much more efficiently than current methods and also designs strong and stable binders in wet lab experiments. CloneBO outperforms naive and informed greedy methods as well as LaMBO, a state of the art method for optimizing sequences. In Section 8 we conclude and describe directions for future work.

Our code and model weights are available at `https://github.com/AlanNawzadAmin/CloneBO`.

## 2 RELATED WORK

To iteratively optimize a protein, one can predict sequences using previous measurements (Rapp et al., 2024; Yang et al., 2019; Fannjiang et al., 2022; Brookes et al., 2019). To optimize antibodies for stability in the human body, Hie et al. (2023) and Prihoda et al. (2022) suggest introducing mutations to select for typicality, make them look more typical, measured by the likelihood of a model trained on large databases of protein sequences. More generally, Gruver et al. (2023) and Stanton et al. (2022) avoid suggesting atypical protein sequences by training a latent space to represent a database of sequences and then optimizing in this latent space. However, even the space of typical antibodies is combinatorially large, and therefore challenging to search using only up to thousands of previous measurements. CloneBO builds an informed prior to efficiently search this space.

CloneBO builds this prior using clonal families — sets of sequences evolving to strongly and stably bind a target (Burnett et al., 2018). Biologists infer evolutionary pressures on antibodies by examining individual clonal families (Mascola & Haynes, 2013) or comparing clonal families (Phad et al., 2022). In the lab, "repertoire mining" optimizes antibodies by suggesting mutations from sequences in a clonal family that contains the candidate (Richardson et al., 2021; Olsen et al., 2023). In practice, such a family rarely exists. CloneBO optimizes a candidate by generating new clonal families that contain the candidate and that match experimental data.

To build a prior over measurements in the lab, we assume that sequences in a clonal family are distributed with abundance according to their fitness, and that fitness is close to the function we measure in the lab. These are standard assumptions in generative modelling of protein sequences (Weinstein et al., 2022) — one fits a distribution $p$ to a set of protein sequences, then uses $\log p(X)$ as an estimate of the fitness of a sequence $X$; this fitness then correlates strongly with the function of the protein as measured in the lab (Frazer et al., 2021; Riesselman et al., 2018; Notin et al., 2022; Shin et al., 2021). In our case, each clonal family has its own fitness function which we use to build a prior over fitness functions. Our model, CloneLM, models clonal families as sets of sequences, similar to the architecture of Truong & Bepler (2023) who used a language model to model protein families as sets of sequences.

For each clonal family we observe a set of sequences which evolve with respect to a latent fitness function drawn from a prior. Instead of attempting to build an explicit latent variable model, Fong et al. (2024) suggest performing Bayesian inference with a "martingale posterior". Instead of sampling and conditioning on the latent variable, they do the same with a large number of observations. Lee et al. (2022) suggests using this approach for Bayesian optimization. Falck et al. (2024) suggests that, with some bias, large language models can perform martingale posterior inference. We take this approach when sampling from our prior. We use a large language model to flexibly fit observed sequences and sample sets of sequences, i.e. clonal families, as draws from our prior.

When proposing sequences, we sample a clonal family from an autoregressive language model, CloneLM, but condition its output to fit experimental measurements. To do so, we build a twisted sequential Monte Carlo procedure (Whiteley & Lee, 2014) in which we bias the generation of each letter towards the posterior. This technique is used to sample from filtering models (Lawson et al., 2023; 2024), large language models (Zhao et al., 2024), or diffusion models (Trippe et al., 2023).

Complementary with work on iterative design are structure-based *de novo* design methods which aim to predict antibody sequences that bind a particular antigen (Jin et al., 2021; Luo et al., 2022; Kong et al., 2023). These models have the potential to design starting sequences for iterative optimization. These models could in principle also be used for iterative design, but cannot make use of a pool of previous measurements, and must have access to structure. We show below empirically that these models are not well suited for this task.

## 3  BACKGROUND

We start with an antibody variable domain $X_0$, a sequence of $110 \sim 130$ letters made of the 20 amino acids, identified to bind a target of interest. $X_0$ often does not bind the target strongly enough or is unstable in the human body, making it unsuitable as a therapeutic. We therefore iteratively propose sequences we expect are stronger or more stable binders, $\hat{X}_1, \ldots, \hat{X}_N$, and measure their binding or stability in the lab $Y_1, \ldots, Y_N$.

We assume that our measurements are evaluations of a function $f$ that takes sequences to a scalar measurement of binding or stability in the lab: $Y_n = f(\hat{X}_n)$. To suggest the next sequence, $\hat{X}_{N+1}$, given $\hat{X}_{1:N}, Y_{1:N}$ we can perform Bayesian optimization (Frazier, 2018). First we place a prior on $f$ given our known weak or unstable binder $X_0$, $p(f|X_0)$. Then we infer $f$ by building a posterior, $p(f|X_0, \hat{X}_{1:N}, Y_{1:N})$. Finally, we suggest $\hat{X}_{N+1}$ given our knowledge of $f$, for example by Thompson sampling: we sample a value of $f$ we believe to be plausible, $f \sim p(f|X_0, \hat{X}_{1:N}, Y_{1:N})$, and test the sequence that maximizes this sample, $\hat{X}_{N+1} = \operatorname{argmax}_X f(X)$.

In the lab we often have a limited experimental budget, and therefore want to find a strong or stable binder in as few iterations as possible. To do so, we need an accurate prior on $f$. The ideal prior could in principle be constructed by performing many optimization experiments in the lab for a diverse array of targets and starting candidates and measuring $f$ in each case. Unfortunately, performing a large number of these experiments is prohibitively expensive.

## 4  A PRIOR FROM FITNESS FUNCTIONS OF EVOLVING ANTIBODY SEQUENCES

While we do not have access to a large number of optimization experiments in the lab, we do have access to a large number of similar optimization experiments that occur in our bodies. Our immune system generates antibodies by first identifying candidate sequences that bind a target. It then evolves this sequence towards binding its target more strongly while remaining stable in the body: mutations are introduced to sequences and those sequences with higher "fitness" — those that bind the target more strongly and stably — are selected for reproduction. Each starting candidate

sequence typically produces many diverse sequences that have been evolved to bind a target strongly and stably. For each optimization experiment the immune system performs, we therefore observe a set of evolved sequences $X_1, \ldots, X_M$ known as a "clonal family".

The $f$ we measure in the lab measures binding and stability, similar to a function of the fitness of sequences under selection. Therefore to build a prior over $f$ we start by building a prior over fitness functions $F$. Then in Section 6 we allow for some discrepancy between $f$ and $F$ which may be caused by a difference between measurements in the lab and selection in our bodies.

To get a prior over functions $F$ from observed clonal families, we first note that the distribution of sequences we observe, $p(X_1, X_2, \ldots)$, can be written as a Bayesian model. The probability of observing a set of sequences in a clonal family is *exchangable*, i.e. it does not depend on their order; so, by De Finetti's theorem[1] (Hewitt & Savage, 1955), sequences in each clonal family are generated iid conditional on a latent random variable which we call clone:

$$p(X_{1:M}) = \int \prod_{m=1}^{M} p(X_m|\text{clone})p(\text{clone}).$$

Next we make the standard assumption that families of evolving proteins are distributed with abundance proportional to their fitness (Weinstein et al., 2022), that is,

$$F(X) = \log p(X|\text{clone}). \tag{1}$$

Sella & Hirsh (2005) showed that Eqn. 1 holds exactly if a protein evolves under $F$ over long time scales. In reality, sequences drawn from $p(X|\text{clone})$ can also be correlated by being descendants of the same sequence, but we make the standard assumption that these correlations can be ignored (Weinstein et al., 2022). Finally, we can represent that the initial candidate $X_0$ binds the target by assuming we have observed it in the clonal family, i.e., by looking at $p(\text{clone}|X_0) \propto p(\text{clone})p(X_0|\text{clone})$.

We can therefore sample fitness functions from $p(F|X_0)$ in theory by **1)** sampling clonal families that contain $X_0$, clone$^* \sim p(\text{clone}|X_0)$, and then **2)** we can set $F(X) = \log p(X|\text{clone}^*)$.

## 5 CLONELM: LEARNING A PRIOR OVER FITNESS FUNCTIONS

In this section we fit a model to the distribution of clonal families and use it to sample fitness functions in practice. In principle, we could build a model with an explicit latent variable meant to represent clone. Instead, we take a martingale posterior approach (Fong et al., 2024) — simply by building an accurate model of clonal sequences we learn an implicit prior on clone that we can approximately sample from.

In Section 5.1 we fit an autoregressive large language model, CloneLM, to large scale clonal family data and show it can generate realistic clonal families $X_{1:M}$ that contain a candidate sequence $X_0$. In Section 5.2 we show that given a clone, $X_{0:M}$, CloneLM implicitly infers the fitness function when predicting sequences: $F(X) \approx \log p_{\text{CloneLM}}(X_{M+1}|X_{0:M})$. Finally in Section 5.3 we show CloneLM can therefore sample fitness functions from an implicit prior on clone by generating realistic clonal families that contain $X_0$ then inferring their fitness functions.

### 5.1 FITTING A LARGE LANGUAGE MODEL TO GENERATE CLONAL FAMILIES

We train a large language model on large scale human clonal family data. Each antibody is made up of two amino acid sequences — the "light chain" and the "heavy chain". We train separate models on heavy and light chains of antibody sequences in clonal families.

To build a training set, we collect all sets of human heavy and light chain antibody sequences from the database of Observed Antibody Space (OAS) (Olsen et al., 2022a). We annotate clonal families in each set of sequences using FastBCR (Wang et al., 2023) and remove any clonal family with fewer than 25 sequences. Our dataset contains 908 thousand heavy chain clonal families and 34 thousand light chain clonal families.

We then train autoregressive language models with 377 million parameters based on the Mistral-7B architecture on the heavy and light chain datasets (Jiang et al., 2023). We represent each clonal family as a sequence of tokens made up of all the amino acid sequences in the clonal family each separated by a special sequence-separator token. We place spaces between amino acids so that the tokenizer represents each amino acid with its own token. Our model, CloneLM, accurately fits this

---

[1]We ignore the dependence between the number of sequences we observe, $M$, and the sequences themselves

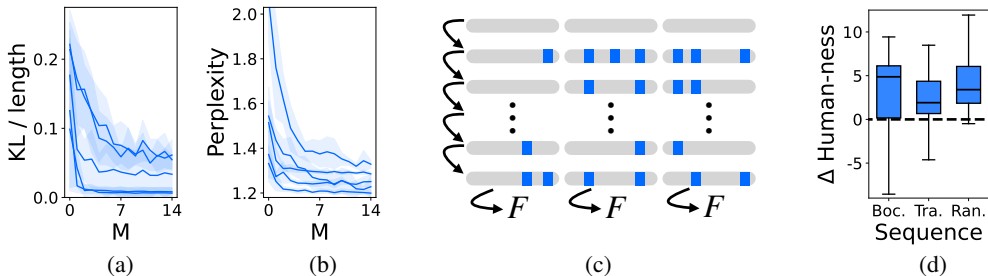

Prompt  QVQLRESGPGLVKPSQTLSLTCTVSGGSFNSGGYYWNRIRQHPGNGLEWIGYMYYSGSTYYNPFIRSRVIISGDTSVNHFSLKLSSVTAADTAVYFCARGYRQSGYSSSWVVDYWGQGTLVNVSS

Clonal family
Sample 1
Sample 2
Sample 3

Figure 2: **CloneLM samples plausible clones.** We compare sequences in a clonal family to families generated by CloneLM conditional on $X_0$ ("Prompt"). We align sequences to $X_0$ and highlight locations where sequences differ from $X_0$ in blue. The sampled clonal families have variants in similar places, are similarly diverse as the real one, and share similar variants within each family.

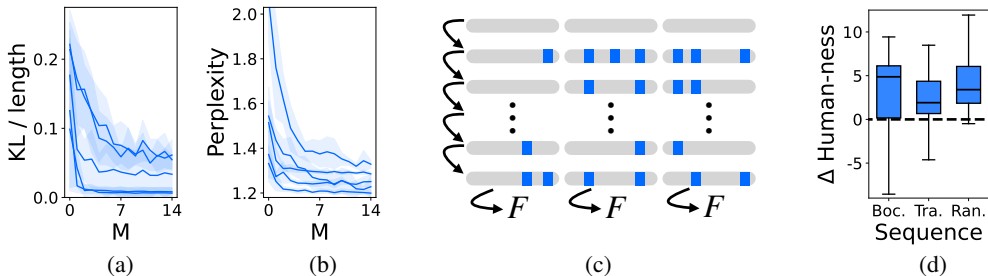

Figure 3: **CloneLM is a prior over fitness functions.** (a) For 5 different clonal families, with starting sequences $X_0$, $p_{\text{CloneLM}}(X_{M+1}|X'_{1:M})$ gets close to $p_{\text{CloneLM}}(X_{M_{\text{large}}+1}|X_{1:M_{\text{large}}})$ in KL. We shade one standard deviation across 10 samples of $X_{1:M_{\text{large}}}, X'_{1:M}$. (b) For 5 different heavy chain clonal families, $p_{\text{CloneLM}}(X|X_{0:M})$ better predicts sequences in a clonal family when conditioned on more sequences from that same clonal family $X_{0:M}$. We shade one standard deviation across 10 samples of $X_{1:M}$. (c) To sample from our prior $F \mid X_0$ we perform a martingale posterior procedure. (d) We evolve three antibody therapeutics with three mutations from 25 sampled fitness functions. These sequences evolve to look more like human antibodies.

data — it achieves a test perplexity of 1.276 on the heavy chain data and 1.267 on the light chain data. We provide details of the data curation and of training the models in App. A.1.

To see if CloneLM generates realistic clonal families, in Fig. 2 we compare a heavy chain clonal family from the test set to clonal families generated by CloneLM conditional on a randomly selected sequence from the original family $X_0$. We see the sampled clonal families are similarly diverse to the real clonal family, include variants in similar locations as the true clonal family, and sequences within the same sampled clonal families contain similar variants. We show more examples of generated heavy and light chain clonal families in Appendix C.1.

## 5.2 APPROXIMATELY EXTRACTING A FITNESS LANDSCAPE FROM A CLONAL FAMILY

CloneLM does not explicitly represent the latent variable clone so we cannot exactly query the fitness function $F(X) = \log p(X|\text{clone})$. However, CloneLM approximates the predictive distribution of a Bayesian model $p(X_{M+1}|X_{0:M})$ which implicitly integrates over the latent clone:

$$p(X_{M+1}|X_{0:M}) = \int p(X_{M+1}|\text{clone})dp(\text{clone}|X_{0:M}).$$

As $M \to \infty$, $p(\text{clone}|X_{0:M})$ converges to a point mass at the latent $\text{clone}^*$ generating the sequences. Therefore, in theory, as $M$ becomes large, $\log p_{\text{CloneLM}}(X_{M+1}|X_{0:M})$ should converge to $F$:

$$\log p_{\text{CloneLM}}(X_{M+1}|X_{0:M}) \approx \log p(X_{M+1}|X_{0:M}) \approx \log p(X_{M+1}|\text{clone}^*) = F(X_{M+1}).$$

We see that CloneLM can infer $F$ as such on real data as well — as $M$ becomes large, the predictive of CloneLM, $p_{\text{CloneLM}}(X_{M+1}|X_{1:M})$, approaches convergence and its limit increasingly approaches the distribution of sequences in a clonal family, $p(X|\text{clone})$. First in Fig. 3a we take random sequences $X_{0:M_{\text{large}}}, X'_{0:M}$ from heavy chain clonal families and see if

$p_{\text{CloneLM}}(X_{M+1}|X'_{0:M})$ converges to $p_{\text{CloneLM}}(X_{M_{\text{large}}+1}|X_{0:M_{\text{large}}})$ in Kullback-Leibler divergence. Setting $M_{\text{large}} = 24$, we see that although the divergence does not go to 0, the distributions become very similar as $M$ becomes large. Next in Fig. 3b we take random sequences $X_{0:M}$ from heavy chain clonal families and see if $p_{\text{CloneLM}}(X|X_{0:M})$ approaches $p(X|\text{clone})$. We see indeed that when we use $p_{\text{CloneLM}}(X|X_{0:M})$ to predict sequences in the clonal family, its perplexity is decreasing in $M$.

## 5.3 SAMPLED FITNESS LANDSCAPES AND EVOLVING SEQUENCES

Given our model samples realistic clonal families and can recover $F(X) = \log p(X|\text{clone})$, we can approximately sample from $p_{\text{CloneLM}}(F|X_0)$ using a martingale posterior procedure (Fong et al., 2024) — we sample from the prior of clonal families that contain $X_0$, $X_{1:M} \sim p_{\text{CloneLM}}(X_{1:M})$, and then approximate the fitness function as $F(X) \approx \log p_{\text{CloneLM}}(X|X_{0:M})$ (Fig. 3c).

The fitness functions we sample reflect how our immune systems evolve antibodies. In Fig. 3d we take the heavy chains of three antibody therapeutics, bococizumab, trastuzumab, and ranibizimab, sample fitness functions setting from CloneLM conditional on these sequences with $M = 10$, and iteratively evolve these sequences by adding the most likely mutation under each sampled fitness functions three times. These therapeutics are not originally human sequences and therefore can be unstable in the human body — in particular, bococizumab was discontinued due to harsh side effects. If they were to evolve in our bodies, we would expect them to become more human-like, and therefore likely more stable. Indeed we see that as we evolve these sequences they look more like human antibodies, where human-ness is measured by the likelihood of IgLM (Shuai et al., 2023), a model trained on a large set of human antibodies.

## 6 CLONEBO: INFERENCE WITH EXPERIMENTAL MEASUREMENTS

We now describe how to use our prior over fitness functions $F$ to optimize sequences in the lab. In Section 6.1 we build a prior for measurements in the lab $f$ using our prior for $F$. We cannot exactly condition on the implicit clone, so in Section 6.2 we approximate the posterior over the latent clone with a posterior over concrete clonal families $X_{1:M}$. In Section 6.3 we describe how to sample from our approximate posterior using a twisted sequential Monte Carlo (SMC) procedure. Finally in Section 6.4 we describe how to suggest sequences to perform iterative optimization with CloneLM; we call our method Clone-informed Bayesian Optimization (CloneBO).

### 6.1 BUILDING A PRIOR ON LABORATORY MEASUREMENTS

In our bodies, antibodies are optimized to stably and strongly bind their targets. In principle, we are interested in doing same in the lab. We therefore assume that the function we optimize in the lab, $f$, is approximately drawn from our prior on fitness functions, $F$, while allowing for some discrepancy due to mismatches between measurements in the lab and those in our bodies. For simplicity, we assume that $f$ is an affine linear transformation of $F$ and that the deviation between experiment and fitness is independent normal with error $\sigma^2$; that is, for some $T > 0, C$, calling $F_n = \log p(\hat{X}_n|\text{clone})$,

$$Y_n \mid \text{clone} \sim N(TF_n + C, \sigma^2 I).$$

To reflect our vague uncertainty about $T$ and $C$ we use uniform priors: $C \sim \text{Uniform}(-\infty, \infty)$ and $T \sim \text{Uniform}(0, \infty)$. With these priors, we get an analytical expression for the marginal likelihood.

**Proposition 6.1.** *(Proof in App. D.1.) For some constant $D$, and $R = \sqrt{N}\frac{\text{Std}(Y_{1:N})}{\sigma}\text{Cor}(F_{1:N}, Y_{1:N})$, with $\Phi$ as the Gaussian CDF,*

$$\log p(Y_{1:N}|F_{1:N}) = -\frac{1}{2}\log \text{Cov}(F_{1:N}) + \frac{1}{2}R^2 + \log \Phi(R) + D. \tag{2}$$

The first term in Eqn. 2 pushes the fitness values of the measured sequences, $F_{1:N}$, to be different, while the later terms push $F_{1:N}$ to strongly and positively correlate with $Y_{1:N}$.

As before, we assume we have a starting candidate $X_0$ that belongs in the clonal family. Therefore,

$$p(\text{clone}|Y_{1:N}, \hat{X}_{1:N}, X_0) \propto p(\text{clone})p(X_0|\text{clone})p(Y_{1:N}|F_{1:N})$$

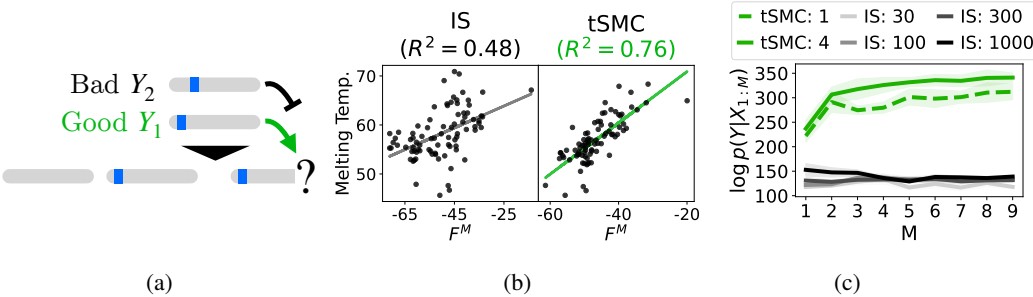

(a)                      (b)                      (c)

Figure 4: **We accurately sample functions from the posterior with a twisted SMC procedure.**
a) To sample from our posterior, we bias our generated sequences to look more like those sequences that were measured in the lab to be good. b) A sample from tSMC better fits the data than an importance sample. We show a line of best fit between $F_{1:N}^M$ (fitness from a clone of $M$ sequences) and $Y_{1:N}$ (measurement) for example clonal families sampled by importance sampling or twisted SMC, with $M = 6$, $D = 300$ particles for IS, and $D = 4$ for twisted SMC. c) We quantify the result from (b) across 10 replicates for various clone sizes $M$.

## 6.2 APPROXIMATING THE POSTERIOR

To infer $f$ given measurements $Y_{1:N}, \hat{X}_{1:N}$, we would like to sample from the posterior $F \sim p(F|Y_{1:N}, \hat{X}_{1:N}, X_0)$. As we only implicitly represent clone, we approximate the posterior by swapping the latent clone for a concrete clonal family, $X_{1:M}$; then, as in Sec. 5.2, we can approximately query the fitness function $F(X) \approx \log p(X|X_{0:M})$.

To create an approximate posterior, we replace $F_n$ with $F_n^M = \log p(\hat{X}_n|X_{0:M})$:

$$\tilde{p}_M(X_{1:M}|Y_{1:N}, \hat{X}_{1:N}, X_0) \propto p(X_{1:M}|X_0)p\left(Y_{1:N}|F_{1:N}^M\right). \tag{3}$$

As $M \to \infty$, $F_{1:N}^M \to F_{1:N}$, so, $\tilde{p}_M$ converges to the distribution one obtains by sampling clone from the posterior and sampling $X_m \sim p(X_m|\text{clone})$ iid:

**Proposition 6.2.** *(Proof in App. D.2) As $M \to \infty$, $\tilde{p}_M$ converges to the true posterior of $X_{0:M}$.*

In practice, we approximate $p(X_{1:M}|X_0)$ and $F_{1:N}^M$ with CloneLM.

## 6.3 SAMPLING FROM THE APPROXIMATE POSTERIOR WITH TWISTED SMC

To sample from $\tilde{p}_M$, we need to generate $M$ sequences from CloneLM such that the probabilities of the $(M + 1)$st sequence matches experimental measurements. Naively, we might sample $X_{1:M}$ from CloneLM and then importance sample. However, the space of sequences is large, and we may fail to resample a mutation that improved measurements in the lab.

Instead we bias generation at each letter by adding measured sequences to our clonal family so that mutations that improve measurements are encouraged and those that harm measurements are avoided (Fig. 4a). We call this bias a "twisted" distribution, a term from the sequential Monte Carlo literature. In practice, this bias does not exactly sample from the posterior, so we efficiently correct for the discrepancy with a sequential Monte Carlo procedure.

**Twisted distributions** Define $X^{(l)}$ to be the letter at the $l$th position of a sequence $X$ and $X^{(:l)}$ as the first $l$ letters in $X$; if $l$ is greater than the length of $X$, we define $X^{(l)}$ as an empty position. To build our twisted distributions, we decompose the likelihood of $\hat{X}_n$ given $M + 1$ sequences into contributions from the first $M$ sequences and contributions from each letter in sequence $M + 1$:

$$
\begin{aligned}
F_n^{M+1} = \log p(\hat{X}_n|X_{0:M+1}) &= \log \frac{p(X_{M+1}|X_{0:M}, \hat{X}_n)}{p(X_{M+1}|X_{0:M})} + \log p(\hat{X}_n|X_{0:M}) \\
&= \sum_l \log \frac{p(X_{M+1}^{(l)}|X_{0:M}, X_{M+1}^{(:l-1)}, \hat{X}_n)}{p(X_{M+1}^{(l)}|X_{0:M}, X_{M+1}^{(:l-1)})} + F_n^M \\
&=: \sum_l F_n^{M+1,(l)} + F_n^M.
\end{aligned}
\tag{4}
$$

We can calculate $F_n^{M+1,(l)}$ by adding $\hat{X}_n$ to the end of the clonal family $X_{1:M}$ and calculating how much the conditional likelihood of the $l$-th letter of $X_{M+1}$ increases. To approximately sample from the posterior therefore, when sampling each letter we bias towards letters with $F_n^{M+1,(l)}$ that better match experimental measurements (Fig. 4a); we call these approximations of the posterior the "twisting distributions" and we define them formally in App. A.2.

The twisted distributions are not the exact marginals. We correct for this discrepancy by sampling $D > 1$ sequences at a time and importance sampling in a sequential Monte Carlo procedure (SMC), which we also describe in App. A.2. The final method is known as twisted SMC, and when $D = 1$, it is equivalent to sampling from the approximations described above.

**Empirical results** We sample a clonal family conditional on laboratory measurements of the melting temperature of 75 related antibodies from an experiment described in Sec. 7. In Fig. 4b and 4c we see that clonal families from twisted SMC fit the experimental data substantially better than clonal families importance sampled from unconditional samples from CloneLM. We also see in Fig. 4c that correcting for bias with $D = 4$ also improves the fit to the data and that as $M$ increases, the likelihood $p(Y_{1:N}|F_{1:N}^M)$ plateaus, reflecting the convergence of $\tilde{p}_M$ to the true posterior. We show similar results for laboratory measurements of binding in App. C.2

### 6.4 BAYESIAN OPTIMIZATION WITH CLONEBO

After sampling $F$ from the posterior we would like to suggest sequences to test in lab. We take a Thompson sampling approach (Russo et al., 2018): we propose testing the sequence predicted to maximize $F(X)$, and therefore maximize $f(X)$, in the lab. We cannot optimize $F(X)$ over all sequences, so in practice we start with 4 sequences with the highest measurements $Y$ and iteratively optimize $F(X)$ over the top substitution for up to 3 substitutions.

In theory, $X_0$ represents a candidate sequence to optimize. In practice, we found it helpful to take a greedy approach — we randomly select $X_0$ from the 4 sequences with the highest measurements $Y$.

Conditioning on a large number of measurements $\hat{X}_{1:N}, Y_{1:N}$ is computationally expensive. To accommodate large $N$, other Bayesian optimization methods build summaries of the measurements, for example by fitting a neural network to them (Stanton et al., 2022). In our case, we only condition on the measurements of sequences predicted to be most informative – we calculate the probability that each $\hat{X}_n$ appears in a clonal family with $X_0$, $p(X_0, \hat{X}_n)$ and condition on the measurements of the 75 most likely sequences. Additional details of CloneBO are provided in App. A.3.

## 7 EXPERIMENTS

Now we demonstrate that CloneBO efficiently optimizes antibody sequences in practice. In Section 7.1 we demonstrate that our method efficiently optimizes fitness functions or laboratory measurements *in silico*. In Section 7.2 we also show that our method suggests mutations that optimize sequences in the lab. We provide experimental details in App. B.

### 7.1 OPTIMIZING ANTIBODIES *in silico*

We show that CloneBO efficiently optimizes fitness functions or measurements in the lab *in silico*. To simulate fitness functions or lab measurements, we train oracles $f$ on real data. We show that CloneBO outperforms naive and informed baselines.

#### 7.1.1 BASELINES

First we consider a naive **Greedy** baseline which suggests a random substitution of one of the top 4 sequences. We also compare to an informed greedy baseline which randomly picks one of the top 4 sequences and introduces the mutation predicted to make the antibody look most like a typical antibody, where typicality in measured by the likelihood of a masked language model trained on antibody sequences, **Sapiens** (Prihoda et al., 2022); this is a popular strategy for making antibodies that are more stable in the body. We also compare to **LaMBO** (Stanton et al., 2022), a state-of-the-art Bayesian optimization method for sequences which builds a latent space using a pool of sequences and fits experimental measurements in this latent space; by conditioning on this latent space, it is less likely to suggest atypical sequences. We also build a LaMBO model informed by the space of antibodies by pretraining its latent space using 100000 antibody sequences from the observed antibody space database (Olsen et al., 2022a), **LaMBO-Ab**. We also compare to other popular

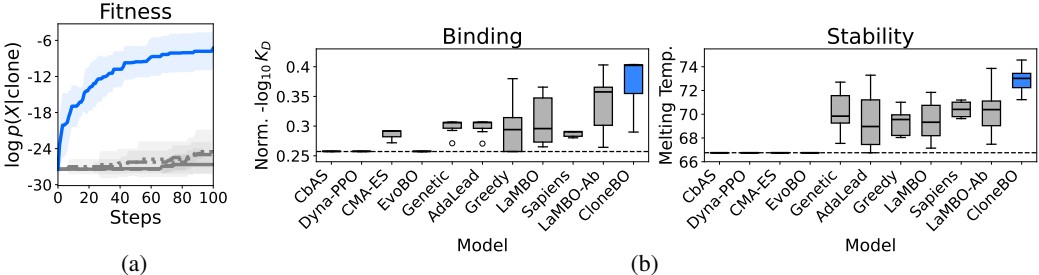

Figure 5: **CloneBO efficiently optimizes antibodies *in silico*.** We show the mean and standard deviation of the best acheived value across 10 replicates. (a) CloneBO efficiently optimizes a fitness function. The blue line is CloneBO; the grey are LaMBO-Ab, LaMBO, Sapiens, and Greedy. (b) CloneBO optimizes binding and stability *in silico* over 100 steps of iterative design (p value is Mann-Whitney). It does significantly better than the next best method for binding (p=0.018 Mann-Whitney) and stability (p=0.006 Mann-Whitney).

strategies for iterative optimization of sequences that do not have antibody-based priors, **Genetic**, **AdaLead** (Sinai et al., 2020), and **EvoBO** (Sinai et al., 2020); **CMA-ES** (Hansen & Ostermeier, 2001), **Dyna-PPO** (Angermueller et al., 2020), and **CbAS** (Brookes et al., 2019). In total, we compare CloneBO to 10 baselines that represent state-of-the-art industry practice[2].

### 7.1.2 RESULTS

**Oracle of a fitness function of a clonal family**    We first demonstrate the potential of the CloneBO prior to accelerate optimization. We build an oracle to simulate the fitness function of a real human clonal family, that is, a function from the CloneBO prior. We trained a language model on a heavy chain clonal family of 10015 sequences from our test set and try to maximize $f(X)$ = the log likelihood of $X$ of this model. We start with a single measurement $\hat{X}_1, Y_1$ where $\hat{X}_1$ is a sequence from the clonal family. Very few mutations of an antibody improve fitness, so in Fig. 5a we see some baselines struggle to identify positive mutations. CloneBO's prior on the other hand gives it knowledge about what sorts of mutations are most likely to improve fitness allowing it to quickly optimize $f$ even at very low $N$.

**Oracles from laboratory measurements of melting temperature and binding**    We next demonstrate the utility of CloneBO in an *in silico* simulation of a realistic in-lab setting. We trained oracles on lab measurements from an experiment that aimed to optimize a VHH domain, a sequence similar to an antibody heavy chain, for binding and stability at high temperature. We trained neural network ensembles on the melting temperature and binding (measured in $-\log K_D$) measurements and try to maximize $f(X)$ = the mean predictions of these ensembles. We simulate starting part way through this experiment by starting with 1000 measurements $\hat{X}_{1:1000}, Y_{1:1000}$ where $\hat{X}_{1:1000}$ are the first 1000 sequences measured in the experiment and $Y_{1:1000}$ are oracle predictions. We do not expect mutations outside of the CDR regions of an antibody to substantially affect binding, so we only allow mutations in these regions of the sequence when optimizing for binding. In Fig. 5b, we see that after 100 steps, greedy methods and methods informed by an antibody prior optimize antibodies more efficiently than previous methods against these oracles; in particular, CloneBO outperforms all baselines. In App. C.3 we plot these results against $N$ and in table form.

**Comparison to structure-based design model for binding SARS CoV.**    In App. C.4 we show that CloneBO can also efficiently optimize antibodies *in silico* for SARS CoV binding as measured by a predictor trained on CoVAbDaB (Raybould et al., 2020). In particular we see CloneBO beats structure-based design method DiffAb (Luo et al., 2022).

**Ablations and sensitivity**    In App. C.5 we show that CloneBO accelerates optimization by building an accurate posterior – the performance of CloneBO is harmed when we ablate sampling large clonal families, conditioning on experimental data, or our twisted SMC sampling strategy. We also

---

[2]Note that some of these methods are developed for different regimens, such as short sequences or large amounts of training data. Their performance here does not necessarily represent their performance for the problems they are optimized for.

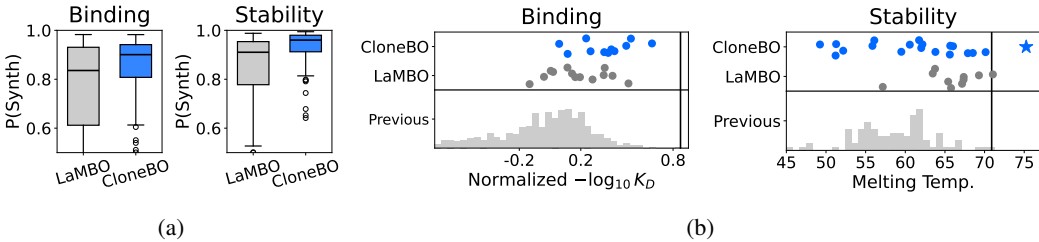

Figure 6: **CloneBO efficiently optimizes antibodies *in vitro*.** *LaMBO* in this plot refers to *LaMBO-Ab*. (a) CloneBO design sequences predicted to be synthesizable. (b) CloneBO designs strong and stable binders in the lab. Measurements from previous rounds are shown in a histogram. The vertical black line represents the best value previously achieved.

perform two other ablations demonstrating that our results above are reliable – we show 1) CloneBO is robust to different starting sequences and starting pool sizes, and 2) CloneBO can efficiently optimize antibodies when $f$ deviates from the CloneBO prior, especially at low $N$. Finally we sweep hyperparameters to show that CloneBO is not particularly sensitive to hyperparameters other than the amount of noise in the data; we describe a heuristic that allows us to make a good choice for the amount of noise.

### 7.2    OPTIMIZING AN ANTIBODY *in vitro*

We now demonstrate that CloneBO can design sequences as part of a real-world antibody optimization campaign. We started with 1000 lab measurements of binding and melting temperature[3] (visualized in Fig. 7) and designed 200 sequences using CloneBO and our strongest baseline, LaMBO-Ab, for one wetlab iteration of optimizing binding or melting temperature. In a real world optimization campaign, this one step would be repeated many times.

Before measuring designed sequences, sequences need to be synthesized; sequences which are atypical can fail to synthesize, making their measurement impossible. In Fig. 6a we plot the predicted synthesizability of sequences from CloneBO and LaMBO-Ab; sequences from CloneBO are significantly more synthesizable (Mann-Whitney $p < 1e-5$), suggesting they are more realistic. We next measure 20 sequences designed by CloneBO and LaMBO-Ab that are predicted to synthesize.

In Fig. 6b we plot sequences we were able to measure; we include any sequences that were proposed that we had measured in previous experiments. We see that sequences from CloneBO achieve the best binding and stability. Our strongest binder is only beaten by 2 / 997 previously measured sequences and our most stable sequence beats the previously measured sequences by a large margin. We also conclude that sequences from CloneBO are significantly stronger binders than those from LamBO-Ab (Mann-Whitney $p = 0.021$). We discuss these results in more detail in App. C.6.

## 8    CONCLUSION

To develop new disease treatments, antibodies must be optimized for a range of properties. By learning from the human immune system's approach to antibody maturation and therefore substantially accelerating optimization in the lab, CloneBO can help build safer and more effective therapeutics.

An important direction of future work is addressing theoretical and practical limitations of CloneBO. First, CloneBO currently assumes a simple relationship between the fitness of a clonal family and measurements in the lab (Section 6.1). Future work may account for heteroskedasticity or nonlinear relationships. As well, CloneBO evaluates the fitness of a sequence by assessing how likely it is to belong to a clonal family of $X_0$. Future work may attempt to incorporate patterns learned from measurements of diverse sequences which are unlikely to belong to the same clonal family. Finally, the cost of sampling from the CloneBO posterior scales with the number of laboratory measurements $N$ (Section 6.3), so CloneBO scales by conditioning only a subset of measured sequences. Future work could instead build a more scalable model or approximate sampling procedure.

Another important future direction is extending CloneBO to multi-objective iterative Bayesian optimization of antibodies for binding and stability simultaneously. We can do so for example by swapping Thompson sampling for other acquisition functions (Daulton et al., 2020).

---

[3]94 / 1000 sequences were measured for melting temperature and 997 / 1000 were measured for $K_D$.

## REPRODUCIBILITY STATEMENT

We include weights for our CloneBO models and code for implementing sampling in our code release. We describe how to train a CloneBO model, including the parameters that were used to build the training data, in the appendix. Sequences from the iterative optimization experiment are proprietary; we release all other data: we include the trained oracle for Fig. 5a in our code release; we describe how implement baselines and the oracle in Fig. 12 in the appendix.

## ACKNOWLEDGEMENTS

This work is supported by NSF CAREER IIS-2145492, NSF CDS&E-MSS 2134216, NSF HDR-2118310, Capital One, and an Amazon Research Award.

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

# A  DETAILS OF CLONELM AND CLONEBO

## A.1  DATA COLLECTION AND TRAINING CLONELM

**Clonal family data**  We downloaded all data units of human single chain data on OAS (Olsen et al., 2022a). For both light and heavy chain data, we put 10% of these units into a test set, 10% into a validation set, and 80% into a train set. We annotated clonal families in each of these data units with FastBCR (Wang et al., 2023) using the default parameters `cluster_thre = 3, overlap_thre = 0.1, consensus_thre = 0.8`. We removed any clonal families with fewer than 25 sequences. We ended up with 731 thousand heavy chain clonal families for training, 81 thousand for validation, and 96 thousand for testing; and 26 thousand light chain clonal families for training, 4 thousand for validation, and 4 thousand for testing.

**Training CloneLM**  We trained CloneLM using Mistral 7B as our base architecture (Jiang et al., 2023). We scale down the model size to 24 layers of attention blocks with 16 attention heads and a hidden size of 1024 across embedding and all intermediate hidden states. We set our maximum context size to 2048. We end up with a Mistral model containing 377 million parameters. For training, we use a batch size of 2, a gradient accumulation step of 4, and we sweep learning rates over $\{0.0005, 0.00025, 0.0001\}$ using a constant learning rate scheduler. All training is performed on 4 NVIDIA A100-SXM4-80GB GPUs. For human light chain data, we train for 24 hours for 40 epochs. For human heavy chain data, we train for 48 hours for 1 epoch.

## A.2 Twisted Sequential Monte Carlo

**Twisted distributions**  We approximate the marginal $\tilde{p}_{M+1}(X_{M+1}^{(:l)}, X_{0:M})$ just as in Section 6.2 by replacing $F_{1:N}^{M+1}$ in Eqn. 3 with $F_{1:N}^{M+1,(:l)} = \sum_{l'=1}^{l} F_n^{M+1,(l')} + F_n^M$,

$$\tilde{p}_{M+1}^{(:l)}(X_{M+1}^{(:l)}, X_{0:M}) \propto p(X_{1:M}, X_{M+1}^{(:l)}|X_0)p(Y_{1:N}|F_{1:N}^{M+1,(:l)}).$$

We call this approximation the "twisted" distribution.

If we pretend that these twisted distributions are exact marginals and that $\tilde{p}_M$ is also the marginal of $\tilde{p}_{M+1}$, we can therefore sample each sequence letter-by-letter according to

$$X_{M+1}^{(l+1)} \sim \tilde{p}_{M+1}^{(:l+1)}(X_{M+1}^{(l+1)}|X_{M+1}^{(:l)}, X_{0:M}) \propto p(X_{M+1}^{(l+1)}|X_{M+1}^{(:l)}, X_{0:M})p(Y_{1:N}|F_{1:N}^{M+1,(:l+1)}). \quad (5)$$

The first term in Eqn. 5 samples the next letter according to the unconditional distribution. The second term is a bias that upweights letters $X_{d,M+1}^{(l+1)}$ for which $F_{1:N}^{M+1,(:l+1)}$ correlates with $Y_{1:N}$; this usually means upweighting letters that are more likely if sequences that were measured to have high $Y_n$, $\hat{X}_n$, were included in the clonal family.

**Importance sampling**  The twisted distributions are not exactly the marginals. We can correct for this discrepancy with sequential weighted importance sampling. Say we have $X_{0:M}, X_{M+1}^{(:l)}$ approximately sampled from $\tilde{p}_{M+1}^{(:l)}(X_{0:M}, X_{M+1}^{(:l)})$ with importance weight $w^{M+1,(:l)}$. Then we can calculate the importance weight of $X_{0:M}, X_{M+1}^{(:l+1)}$ by multiplying by the ratio between $\tilde{p}_{M+1}^{(:l+1)}(X_{0:M}, X_{M+1}^{(:l+1)})$ and $\tilde{p}_{M+1}^{(:l+1)}(X_{M+1}^{(l+1)}|X_{M+1}^{(:l)}, X_{0:M})\tilde{p}_{M+1}^{(:l)}(X_{0:M}, X_{M+1}^{(:l)})$, so,

$$\frac{w^{M+1,(:l+1)}}{w^{M+1,(:l)}} \propto \frac{p(X_{M+1}^{(l+1)}|X_{M+1}^{(:l)}, X_{0:M})}{\tilde{p}_{M+1}^{(:l+1)}(X_{M+1}^{(l+1)}|X_{M+1}^{(:l)}, X_{0:M})} \times \frac{p(Y_{1:N}|F_{1:N}^{M+1,(:l+1)})}{p(Y_{1:N}|F_{1:N}^{M+1,(:l)})}.$$

Therefore if we have $D$ samples $X_{0:M+1}^{1:D}$ with weights $w_{1:D}^{M+1}$ then we can approximately sample from $\tilde{p}_{M+1}$ by sampling $X_{0:M+1}^d$ with probability $\tilde{w}_d^{M+1} = \frac{w_d^{M+1}}{\sum_{d'} w_{d'}^{M+1}}$.

**Sequential Monte Carlo**  Say we are iteratively sampling and weighting $D$ sets of sequences and the importance weight for one set $w_d^{M,(:l)}$ becomes much smaller than that of the others. Ideally we wouldn't waste any more compute on sampling the rest of the sequence. This is the idea of sequential Monte Carlo — while generating each set of sequence letter-by-letter, every so often, we resample the sets of sequences with probabilities $\tilde{w}_{1:D}^{M,(:l)}$. To decide when to resample, we calculate the essential sample size $\sum_d (\tilde{w}_d^{M,(:l)})^2$ and resample when it goes below $\sqrt{D}$, a classic heuristic. After we resample, we reset the weights $w_d = 1/D$. As $D \to \infty$ we expect to approximate $\tilde{p}_{M+1}$ arbitrarily well.

We also note when using the predictive distributions of CloneLM, Eqn. 4 is an approximation. The discrepancy comes from the fact that $F_n^{M+1}$ is the conditional probability of $\hat{X}_n$ as the $M + $2nd sequence while $\tilde{F}_n^{M+1} := \sum_l F_n^{M+1,(l)} + F_n^M$ is the probability of $\hat{X}_n$ as the $M + $1st sequence. For $p$, these probabilities are identical, but this may not be exactly the case for CloneLM. Therefore, once we have sampled all the letters in a sequence $X_M$ then we have sampled from a distribution proportional to

$$p(X_{0:M})p(Y_{1:N}|\tilde{F}_{1:N}^M).$$

Therefore we also resample at this stage after multiplying the importance weight of sample $d$, $w_d^M$ by

$$\frac{p(Y_{1:N}|F_{1:N}^M)}{p(Y_{1:N}|\tilde{F}_{1:N}^M)}.$$

## A.3 Experimental Details of CloneBO

Before conditional generation, we normalized $Y_{1:N}$ to to $\tilde{Y}_n = (Y_n - \text{startmean})/\text{startstd}$ where startmean and startstd are the mean and standard deviation of the initial dataset $Y_{1:N}$. In our

experiments we used $D = 4$ during twisted SMC, and generated clones of size $M = 6$. We tempered $\sigma$ by the maximum number of sequences we conditioned on, i.e. we used $\sigma = \tilde{\sigma}/\sqrt{75}$ where $\tilde{\sigma} = 0.25$. We run each experiment on a single NVIDIA A100 GPU with 80GB of memory; 100 steps with CloneBO takes roughly 10 hours.

# B  EXPERIMENTAL DETAILS

## B.1  BASELINES

We implemented Sapiens using the code in `https://github.com/Merck/Sapiens` under the MIT licence. We suggested mutations to a sequence by taking the highest likelihood mutation suggested by `sapiens.predict_scores` that had not previously been measured.

We implemented LaMBO using the code in `https://github.com/samuelstanton/lambo` under the Apache-2.0 licence. We used default hyperparameters for the masked language model version of LaMBO. To restrict mutations to the CDRs, we kept sampling mutations from LaMBO until only the CDR was modified.

To build LaMBO-Ab we pretrained the latent space of a LaMBO MLM model on a training set of 100000 antibody sequences. We built the training set by taking one sequence from each of 100000 random clonal families from the CloneLM training set.

We also compare to two other genetic algorithms with trained surrogates, AdaLead and Genetic; a NN ensemble Bayesopt method, "Evo-BO"; an evolution method, CMA-ES (Hansen & Ostermeier, 2001); an RL method, Dyna-PPO (Angermueller et al., 2020); and an adaptive sampling method, CbAS (Brookes et al., 2019). The first three methods are described in Sinai et al. (2020). We implemented these methods with code from FLEXS (Sinai et al., 2020) using the code from `https://github.com/samsinai/FLEXS/tree/master` under an Apache-2.0 license; we used default settings for all methods.

## B.2  TRAINING ORACLES AND INITIALIZING OPTIMIZATION

**Oracle of a fitness function of a clonal family**  We trained an oracle language model adapted from Llama2 (Touvron et al., 2023) on a single reference human heavy chain clone. There are in total 10015 sequences in the clone and we split them into 90% train, 5% validation, and 5% test sets. Due to the scarcity of our data, we downscaled a Llama2 model with 7 billion parameters by keeping 12 hidden layers with hidden state size of 512. We used 4 attention heads, 4 key-value heads, and kept the context size at 2048. We ended up with a language model containing 50 million parameters. We trained our model on a single NVIDIA A100-SXM4-80GB GPU for 10 epochs with a batch size of 32, a gradient accumulation step of 2, a learning rate of 0.0005. We obtained a perplexity value of 1.2634 on the validation set. Using the clone oracle, we start optimization with 2 randomly chosen sequences from the test set $\hat{X}_{1:2}$, and predictions from the oracle $Y_{1:2}$.

**Oracles from lab measurements of therapeutic Antibodies**  We were provided temperature and binding measurements of 6880 sequences from an iterative optimization experiment performed in the lab. We aligned these sequences to a reference antibody sequence and trained ensembles of 10 CARP/Bytenet models (Kalchbrenner et al., 2016; Yang et al., 2022) on the one hot encodings of the aligned sequences to fit the temperature and binding data. The $K_d$ model ensemble had a measured vs. predicted spearman correlation in crossvalidation of 0.95, while the the $T_m$ model ensemble had an spearman correlation of 0.72. We use the mean prediction of the ensemble as $f$. We start optimization given the measurements of the first 1000 sequences of this experiment $\hat{X}_{1:1000}$, and predictions from the oracle $Y_{1:1000}$.

### B.3 TRAINING DATA FROM ITERATIVE OPTIMIZATION

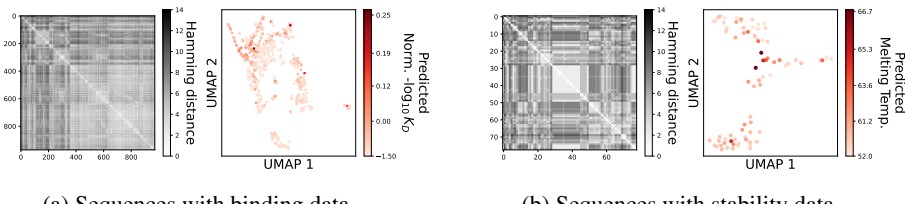

(a) Sequences with binding data.     (b) Sequences with stability data.

Figure 7: Starting pool for (a) binding and (b) stability optimization. We plot the Hamming distance matrices and UMAPs. sequences are coloured by predicted property

### B.4 LAB VALIDATION

We built a predictor of synthesizability from measures of expression just the same as predictors of melting temperature and binding in Sec B.2. The predictor achieves a test AUROC of 0.87.

Designed sequences were synthesized via cell-free protein synthesis (Dopp & Reuel, 2020) in 96-well format and purified via Protein A binding on Pierce magnetic beads. Purity and yield were confirmed before further analysis. Affinities ($-\log K_D$) were measured with Bio-Layer Interferometry (BLI) on an Octet instrument, with the antigen immobilized at three different dilutions of antibody. Thermostability (melting temperature) was measured by Nano differential scanning interferometry (NanoDSF) on an Uncle instrument.

## C SUPPLEMENTARY RESULTS

### C.1 MORE EXAMPLE GENERATED CLONAL FAMILIES

In this section we show more light and heavy chain clonal families generated from CloneLM. In Fig. 8b and Fig.s 9b, 9c, we see that sequences generated by CloneLM include can introduce insertions and deletions. The large sets of deletions in the sequences of the clonal family in Fig. 8b are due to the fact that some sequences in OAS are missing the beginning or end of their sequences (Olsen et al., 2022b).

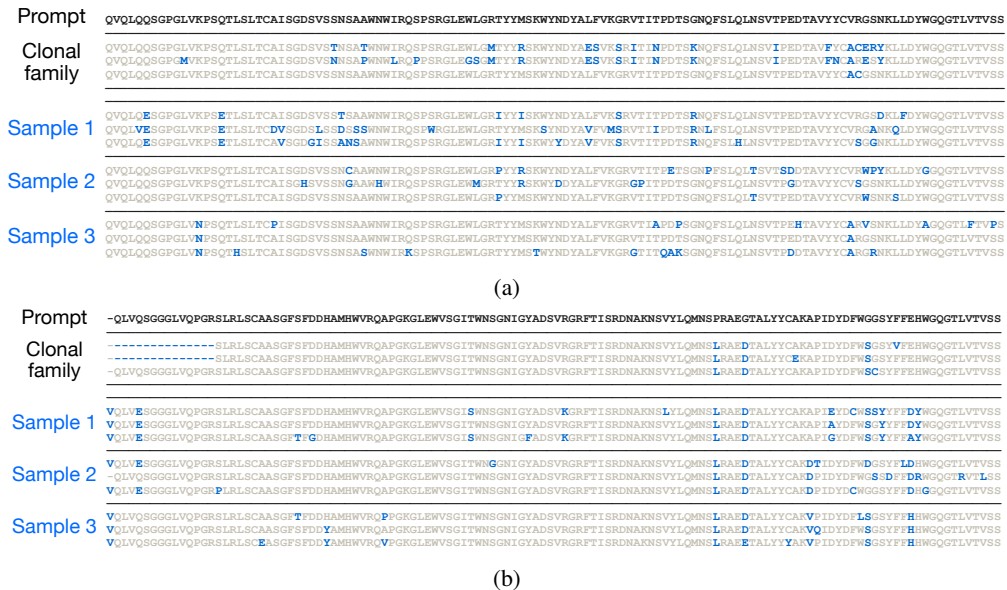

Figure 8: **Examples of heavy chain clonal families generated by CloneLM.**

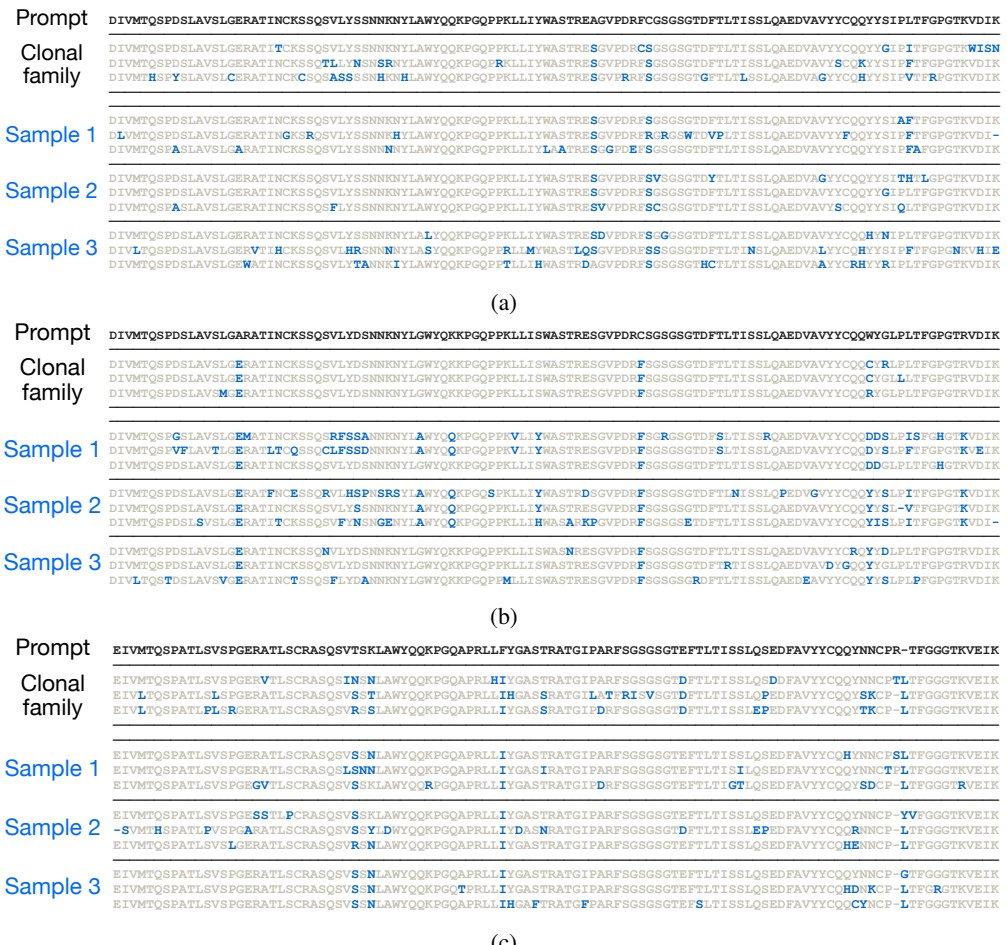

Figure 9: **Examples of light chain clonal families generated by CloneLM.**

## C.2 TWISTED SMC FITTING AFFINITY DATA

In Fig. 10 we show similar results to Fig. 4c for 75 laboratory measurements of binding.

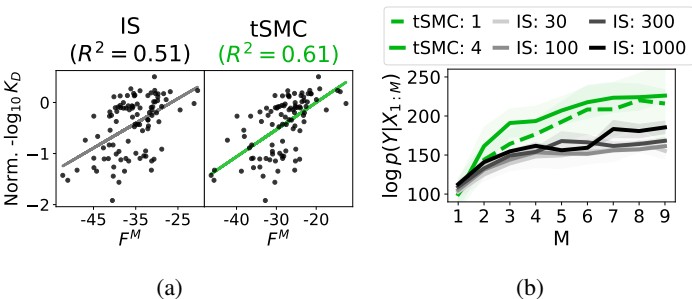

Figure 10: **Twisted SMC fits laboratory measurements of affinity.** Experiments are similar to those in Fig. 4c.

## C.3 Efficient optimization versus number of steps

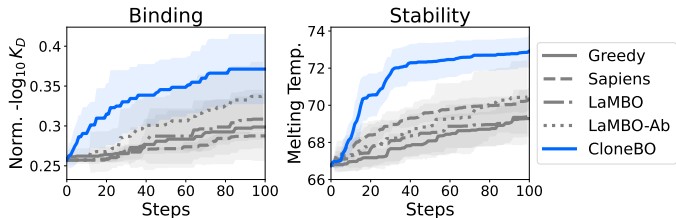

Figure 11: Results of Fig. 5b for various $N$ for representative baselines.

| Method | $\log p(X|\text{clone})$ | Norm. $\log_{10} K_D$ | Melting temp. |
|---|---|---|---|
| CbAS | -27.42 | 0.2572 | 66.76 |
| Dyna-PPO | -27.42 | 0.2572 | 66.76 |
| CMA-ES | -27.42 | 0.2853 | 66.76 |
| EvoBO | -27.42 | 0.2572 | 66.76 |
| Genetic | -25.94 | 0.3001 | 70.19 |
| AdaLead | -25.30 | 0.2998 | 69.42 |
| Greedy | -26.65 | 0.2986 | 69.28 |
| LaMBO | -24.99 | 0.3085 | 69.40 |
| Sapiens | -24.51 | 0.2875 | 70.39 |
| LaMBO-Ab | -24.51 | 0.3381 | 70.44 |
| **CloneBO** | **-7.29** | **0.3713** | **72.92** |

Table 1: Result from Fig. 5 in table form. We report the mean best value achieved across 10 replicates.

## C.4 Optimizing antibodies to bind SARS CoV

Above, we validated CloneBO using predictors trained on large-scale mutational data from a lab as well as in *in vitro* experiments – these are the most reliable evaluations of CloneBO. As another evaluation of CloneBO, and to compare it to structure-based design models, here we explore optimizing the CDRH3s of antibodies for binding SARS CoV 1 and 2 in humans as measured by predictors used in Jin et al. (2021) (downloaded from `https://github.com/wengong-jin/RefineGNN`). We caution however that while we expect CloneBO to be a good prior for SARS binding, these predictors have large epistemic uncertainty as they are trained on only a few thousand extremely diverse sequences. We thus expect we are optimizing an objective similar to those in Fig. 14 so that CloneBO should perform best at low N.

These predictors were trained on only functional antibodies and may not generalize outside this set. So, we optimize binding + humanness (measured by IgLM likelihood); we standardize both binding and humanness to the same variance.

We start with $N = 1$ sequence from CoVAbDab. In Fig. 12 we see that for 6 randomly selected starting sequences, CloneBO is consistently among the most efficient methods for CoV1 and, at low N, for CoV2 as well.

We were interested in seeing if structure-based design methods could also be used for iterative design. When structure is available, we therefore also compare to a state-of-the-art structure method, DiffAb (Luo et al., 2022). To perform iterative design with DiffAb, we greedily pick one of the 4 best measured sequences and optimize it as in Sec 4.3 of Luo et al. (2022). We see that CloneBO beats this algorithm.

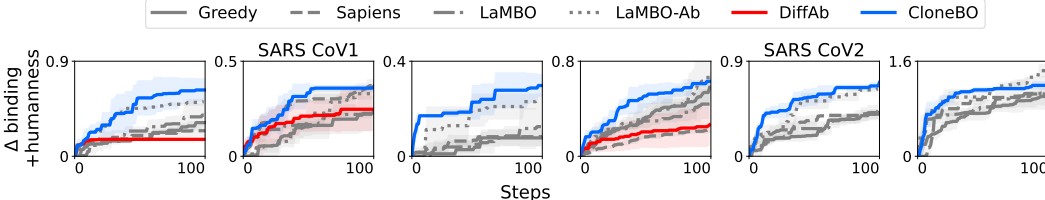

Figure 12: CloneBO efficiently optimizes 6 sequences from CovAbDab for binding to SARS CoV1 (first 3 columns) or CoV2 (last 3 columns). In particular it outperforms a structure-based baseline (DiffAb) for the 3 sequences with an available structure. We show mean and standard deviation achieved across 3 replicates.

## C.5 ABLATIONS OF *in silico* OPTIMIZATION

**CloneBO efficiently optimizes sequences by building an accurate posterior.** We now show that CloneBO efficiently optimizes sequences by conditioning on experimental measurements and accurately sampling from the martingale posterior. In Fig. 13 we see optimization is often harmed by 1) not accurately sampling from the martingale posterior by only sampling clones of size $M = 1$, 2) using a naive importance sampling procedure instead of twisted SMC, or 3) not conditioning on previous measurements. We see the $M = 1$ or importance sampling ablations have less of an effect when optimizing fitness, potentially because it is easier to condition on data from a function from the CloneBO prior.

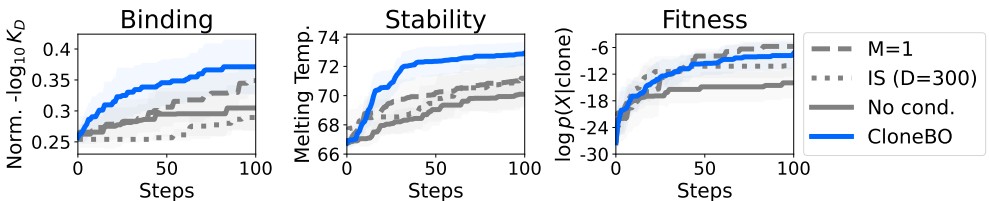

Figure 13: **CloneBO better optimizes antibodies than models that ablate accurately sampling from the posterior.** We shade a standard deviation across 3 replicates. We show results for optimizing binding *in silico*, stability *in silico*, and the fitness function of a clone.

**CloneBO is robust to different starting pool sizes and deviations from its prior at low N.** In Fig. 14a we optimized for stability as in Fig. 5b of the paper with a starting pool of various $N$ measurements. We see that CloneBO outperforms our baselines across these different starting stabilities and starting pool sizes, especially at low $N$.

In Fig. 5a in the paper we showed that when optimizing an objective from CloneBO's prior (the fitness function of a clone), $F$, CloneBO strongly outperforms baselines. Here we mix $F$ with a random neural network, $G$, and optimize $wF + (1 - w)G$; $w \in [0, 1]$ controls how well the CloneBO prior describes the objective. We start with $N = 2$ sequences. In Fig. 14b we see that in this small $N$ setting, CloneBO outperforms other methods even when the objective only somewhat matches the prior ($w = 0.4$). Even at $w = 0.2$, CloneBO is the best method at very low N (up to N=25).

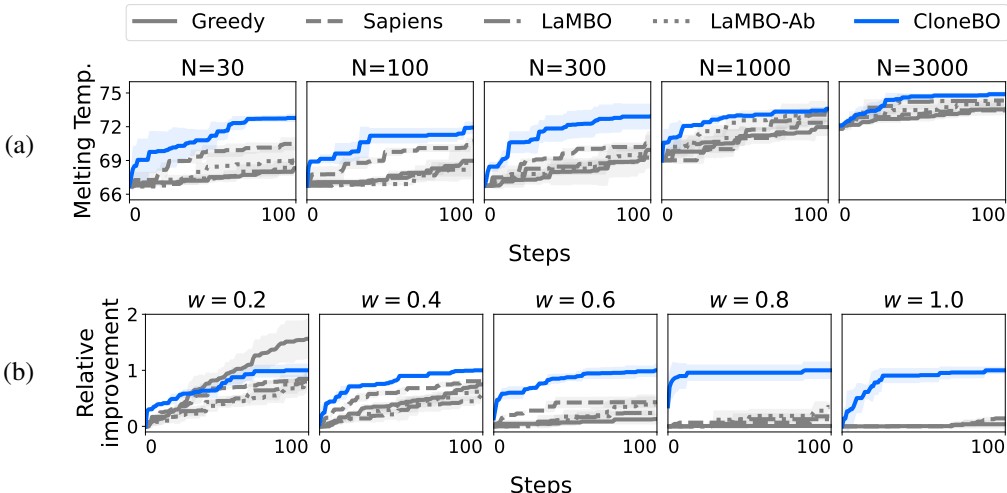

Figure 14: **CloneBO is robust** to ablating (a) the starting pool size or (b) how well its prior describes the objective. We show the mean and standard deviations of the best achieved value across 3 replicates.

**Sensitivity to hyperparameters *in silico***   We now investigate the sensitivity of our results to 3 hyperparameters: the noise level in the likelihood $\tilde{\sigma}$, the maximum number of allowed mutations $L$, and the number of sequences we draw $X_0$ from $K$, as described in Sec. 6.4. In our experiments above we use $\log_2 \tilde{\sigma} = -2$, $L = 3$, and $K = 4$.

We picked $K$ and $L$ based on intuition. We picked $\tilde{\sigma}$ by looking at the clones generated when conditioned on the initial binding data *in silico* when suing values $\log_2 \tilde{\sigma} = -4, -3, -2, -1$; we noted that when $\log_2 \tilde{\sigma} = -4, -3$, generated clones contained very short or long sequences, indicating it was hard to find clones that fit the data with little noise. Thus we picked $\log_2 \tilde{\sigma} = -2$ to fit the data with as little noise as possible. Performing the same procedure for the *in silico* stability data, we arrived at the same value $\log_2 \tilde{\sigma} = -2$ was a good choice. We fixed this value for all our other experiments.

We see in Fig. 15a that $K = 1$ performs badly when optimizing binding, likely due to a higher chance of getting stuck in a local minima; otherwise CloneBO is not very sensitive to $K$. In Fig. 15b we see CloneBO is also not very sensitive to $L$. In Fig. 15c we note CloneBO is sensitive to the choice of $\tilde{\sigma}$ but our procedure described above picked the optimal $\tilde{\sigma}$ for binding and stability. For fitness, we noted the posterior is easy to sample from even at $\log_2 \tilde{\sigma} = -3$ (we see this manifest in Fig. 13 as well) and indeed a smaller value of $\tilde{\sigma}$ is optimal. This suggests one can optimize antibodies more efficiently by improving the CloneBO likelihood, potentially by picking a better $\tilde{\sigma}$ or by adding a prior and marginalizing over it.

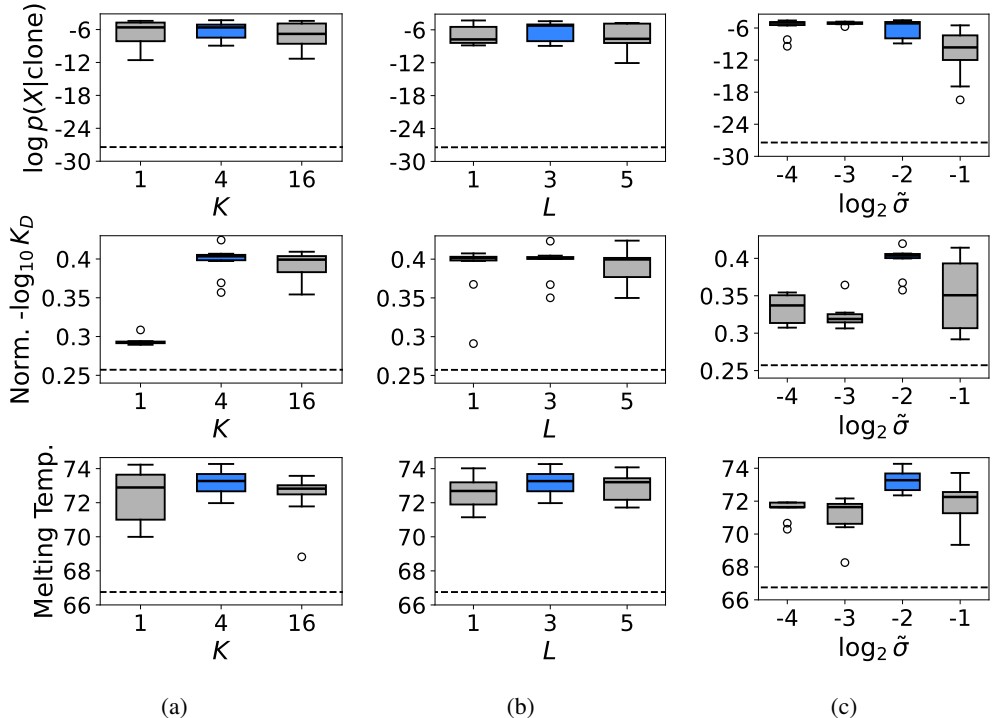

(a)                                (b)                                (c)

Figure 15: **CloneBO sensitivity to hyperparameters for fitness, binding and stability.** Experiments are run with 10 replicates; blue boxes represent hyperparameters used in the main text . a) Sensitivity to size of pool $X_0$ is randomly selected from, $K$. b) Sensitivity to maximum allowed number of mutations, $L$. c) sensitivity to noise in likelihood, $\tilde{\sigma}$.

### C.6    ADDITIONAL DISCUSSION OF *in vitro* RESULTS

We were able to measure the affinities of 9 sequences for CloneBO, and 11 for LaMBO-Ab, and the melting temperatures of 19 sequences from CloneBO and 10 sequences from LaMBO-Ab. Adding in previously measured sequences, we were able to get two more affinity measurements for CloneBO and 2 more for LaMBO-Ab, and no other affinity measurements. Note when interpreting these results that affinity and stability may be correlated with dropout and whether or not a sequence was previously measured. In Fig. 16 we plot the affinity measurements removing the previously measured sequences; the results are qualitatively similar to those of Fig. 6.

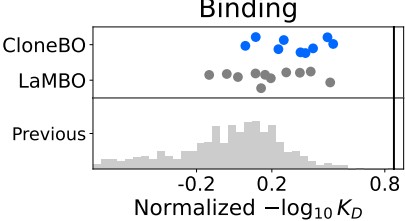

Figure 16: The result of Fig. 6b for affinity with only sequences that were newly measured.

# D  THEORETICAL RESULTS

## D.1  ANALYTIC FORM OF MARGINAL LIKELIHOOD

**Proposition D.1.** *(Proof of Prop. 6.1) For some constant $C$,*

$$\log p(Y_{1:N}|F_{1:N}) = -\frac{1}{2}\log \text{Cov}(F_{1:N}) + \frac{1}{2}R^2 + \log\Phi(R) + C'$$

*Proof.* We first put a wide prior on $M$, $M \sim N(0, \tau^2)$, and then later send $\tau \to \infty$. Then we can marginalize $M$ out to get

$$Y_n \sim N(\beta F_n, \sigma^2 I + n\tau^2 e \otimes e)$$

where we define $e$ to be the vector with $1/\sqrt{N}$ in each position, $e = \vec{1}/\sqrt{N}$. Calling $\Sigma = \sigma^2 I + n\tau^2 e \otimes e$, $\sigma_\beta^2 = (F_{1:N}^T \Sigma^{-1} F_{1:N})^{-1}$, $\mu_\beta = \sigma_\beta^2 F_{1:N}^T \Sigma^{-1} Y_{1:N}$, we get

$$\int_0^\infty p(Y_{1:N}, \beta | F_{1:N}) d\beta \propto \int_0^\infty e^{\beta F_{1:N}^T \Sigma^{-1} Y_{1:N} - \frac{1}{2}\beta^2 F_{1:N}^T \Sigma^{-1} F_{1:N}} d\beta$$

$$= (2\pi\sigma_\beta^2)^{1/2} e^{\mu_\beta^2/2\sigma_\beta^2} P(N(\mu_\beta, \sigma_\beta^2) > 0)$$

$$\propto \sigma_\beta e^{\mu_\beta^2/2\sigma_\beta^2} \Phi\left(\frac{\mu_\beta}{\sigma_\beta}\right).$$

Now,

$$\Sigma^{-1} = (\sigma^2(I - e \otimes e) + (\sigma^2 + n\tau^2)e \otimes e)^{-1}$$

$$= \sigma^{-2}(I - e \otimes e) + (\sigma^2 + n\tau^2)^{-1} e \otimes e$$

$$\to \sigma^{-2}(I - e \otimes e) \text{ as } \tau \to \infty$$

$$= N\sigma^{-2}\left(\frac{1}{N}I - \left(\frac{1}{N}\vec{1}\right) \otimes \left(\frac{1}{N}\vec{1}\right)\right).$$

Therefore, as $\tau \to \infty$, $F_{1:N}^T \Sigma^{-1} F_{1:N}$ is $N\sigma^{-2}$ times the variation of $F_{1:N}$, $\text{Var}(F_{1:N})$, and $F_{1:N}^T \Sigma^{-1} Y_{1:N}$ is $N\sigma^{-2}$ times the covariance of $F_{1:N}$ and $Y_{1:N}$, $\text{Cov}(F_{1:N}, Y_{1:N})$.

Therefore, if we call $\frac{\mu_\beta}{\sigma_\beta} \to \sqrt{N}\sigma^{-1}\text{Cor}(F_{1:N}, Y_{1:N})\text{Std}(Y_{1:N}) = R$ then we get that when we send $\tau \to \infty$,

$$\log p(Y_{1:N}|F_{1:N}) = -\frac{1}{2}\log \text{Cov}(F_{1:N}) + \frac{1}{2}R^2 + \log\Phi(R) + C'$$

for some constant $C'$. $\qquad\square$

## D.2  CONVERGENCE OF APPROXIMATE POSTERIOR

We show that the approximate posteriors converge defined in Eqn. 3 converge to the true posterior. We make the mild assumption that the hypothetical latent variable clone has been defined such that clone $\mapsto p(X|\text{clone})$ is measurable and no two variables clone$_1$, clone$_2$ have the same conditional distribution $p(X|\text{clone}_1) \neq p(X|\text{clone}_2)$. We also assume that all measured sequences are antibodies that can plausibly, although maybe with extremely low likelihood, appear in a clone together, i.e. $p(X_0, \hat{X}_{1:N}) \neq 0$. Finally, we assume the sequences $\hat{X}_{1:N}$ are sufficiently diverse so that their log likelihoods cannot all be identical for any clone, i.e. for some $\epsilon > 0$, $\text{Cov}(F_{1:N}) > \epsilon$ with probability 1 under $p(\text{clone}|X_0)$[4].

**Proposition D.2.** *(Proof of Prop. 6.2) Assume* clone $\mapsto p(X|\text{clone})$ *is measurable and injective,* $p(X_0, \hat{X}_{1:N}) \neq 0$, *and* $\text{Cov}(F_{1:N}) > \epsilon$ *or some $\epsilon > 0$ with probability 1 under $p(\text{clone}|X_0)$. Then, as $M \to \infty$, the approximate and true posteriors converge in total variation –*

$$\|\tilde{p}_M(X_{1:M}|Y_{1:N}, \hat{X}_{1:N}, X_0) - p(X_{1:M}|Y_{1:N}, \hat{X}_{1:N}, X_0)\|_{\text{TV}} \to 0.$$

---

[4] We need this assumption to ensure the density $p(Y_{1:N}|F_{1:N}^M)$ is bounded above. Alternatively, one can assume a proper prior on $M$ by picking $\tau$ large but finite in the proof of Prop. D.1.

*Proof.* By our assumptions, by Doob's theorem (Miller, 2018), if clone $\sim p(\text{clone}|X_0)$ and $X_m \sim p(X|\text{clone})$ iid, with probability 1,

$$p(\hat{X}_n|X_{1:M}) \to p(\hat{X}_n|X_{1:M})$$

for each $n$. In particular, we get that $F_{1:N}^M \to F_{1:N}$ and therefore, since, assuming $\text{Cov}(F_{1:N}) > \epsilon$, $p(Y_{1:N}|F_{1:N}^M)$ is a bounded function of $F_{1:N}$,

$$E_{X_{1:M},\text{clone}\sim p(X_{1:M}|\text{clone})p(\text{clone}|X_0)}|p(Y_{1:N}|F_{1:N}^M) - p(Y_{1:N}|F_{1:N})| \to 0.$$

Now we show that the the normalizing constants of the approximate and true posteriors converge.

$$
\begin{aligned}
Z_M := \int p(Y_{1:N}|F_{1:N}^M)dp(X_{1:M}|X_0) &= \int p(Y_{1:N}|F_{1:N}^M)\prod_{m=1}^{M} dp(X_m|\text{clone})dp(\text{clone}|X_0) \\
&= E_{X_{1:M},\text{clone}\sim p(X_{1:M}|\text{clone})p(\text{clone}|X_0)}p(Y_{1:N}|F_{1:N}^M) \quad (6) \\
&\to E_{\text{clone}\sim p(\text{clone}|X_0)}p(Y_{1:N}|F_{1:N}) \\
&= \int p(Y_{1:N}|F_{1:N})dp(\text{clone}|X_0) =: Z
\end{aligned}
$$

By our assumption that $p(X_0, \hat{X}_{1:N}) \neq 0$, for a set of clone of probability greater than 0 under $p(\text{clone}|X_0)$, $p(\hat{X}_n|\text{clone}) > 0$ for all $n$; for this set, $p(Y_{1:N}|F_{1:N}) > 0$ and therefore $Z > 0$.

Now we show that the approximate and true posteriors converge in total variation:

$$
\begin{aligned}
\sum_{X_{1:M}} &\left| \frac{p(Y_{1:N}|F_{1:N}^M)p(X_{1:M}|X_0)}{Z_M} - \frac{\int p(X_{1:M}|\text{clone})p(Y_{1:N}|F_{1:N})dp(\text{clone}|X_0)}{Z} \right| \\
\leq &\sum_{X_{1:M}} p(Y_{1:N}|F_{1:N}^M)p(X_{1:M}|X_0)\left|Z_M^{-1} - Z^{-1}\right| \\
&+ \frac{1}{Z}\sum_{X_{1:M}}\left| \int p(X_{1:M}|\text{clone})\left(p(Y_{1:N}|F_{1:N}^M) - p(Y_{1:N}|F_{1:N})\right)dp(\text{clone}|X_0)\right| \\
\leq &\frac{|Z-Z_M|}{Z} + \frac{1}{Z}E_{p(\text{clone}|X_0)}|p(Y_{1:N}|F_{1:N}) - p(Y_{1:N}|F_{1:N}^M)| \to 0.
\end{aligned}
\quad (7)
$$

$\square$

