# OpenReview forum: "Bayesian Optimization of Antibodies Informed by a Generative Model of Evolving Sequences"
_ICLR.cc/2025/Conference — ICLR 2025 Spotlight_

### Official Review · Reviewer_mvea · 2024-10-30

**Soundness:** 3
**Presentation:** 3
**Contribution:** 3
**Rating:** 8
**Confidence:** 4

**Summary:**

The paper proposes a new way to design antibodies using Bayesian optimisation. In contrast to most closely related existing aproach (Lambo) the authors propose a different way to capture the prior antibody distribution. They train a PLM on clonal families, to capture how the imune system diversifies antibodies. They also introduce a twisted sequential Monte Carlo procedure to better incorporate previous lab measurements (sucessful designs). The in silico and in vitro results show that the proposed procedure improves upon Lambo in terms of antibody synthesizeability and stability.

**Strengths:**

I find it quite a unique and nice idea to try to produce local diversification around the seed that matches diversification our body produces during imunization and it's implementation is going a bit of the beaten path with for example the martingale posterior. The experiments are quite thorough and convincing, with a strong baseline. The paper itself is well written and easy to follow.

**Weaknesses:**

This is not a big weakness, but I'd say that the CloneLM model makes more Fv mutations than one would expect. Also, in appendix the clonal familly does not fix the missing starting gap '-' in some cases, which is not ideal as it would always be a 'Q' for human Ab with the given subsequent amino acids.

The melting temperature and binding oracle experiment uses VHHs, which I would say is not ideal. While humanized VHHs do look kinda like human VH chain, they are still different, even in trivial ways (e.g. they often have much longer CDR H3). It's nice that the proposed method deals with this distribution shift, but it's not entierly fair to compare to the baselines, that were only developed for 'normal' Abs.

In the paper you write "de novo design method DiffAb", but DiffAb it's not de novo, it needs a co-crystal structure as a starting point.

Figure 4 (b) second plot, only 1 point beats best point of Lambo, the average of Lambo also looks better so I'm not fully convinced that the proposed method produces more stable antibodies. Of course, the KD results are quite convincing.

**Questions:**

When training the ClonalPLM are the clonal families (their order) re-mixed (e.g. for Light chain as it uses many epochs) if not, how is the ordering chosen?

What is the edit distance distribution for the in-vitro designs? In the method description (Section 6.4) you state "iteratively optimize F(X) over the top substitution for up to 3 substitutions", does this mean that designs can only have maximum edit distance of 3 in your experiments? If yes were the baselines constrained to the same edit distance?

---

> ### Author Response · Authors · 2024-11-21
> **Comment 1**
>
> Thank you for your thorough and thoughtful review! We address each of your suggestions and questions below. In addition to these points, we’ve made a number of aesthetic changes to the figures of the paper in the new draft.
>
> >This is not a big weakness, but I'd say that the CloneLM model makes more Fv mutations than one would expect. Also, in appendix the clonal familly does not fix the missing starting gap '-' in some cases, which is not ideal as it would always be a 'Q' for human Ab with the given subsequent amino acids.
>
> Indeed, CloneLM reflects the biases of the data it was trained on, OAS! For example, given the sequencing coverage biases in OAS, CloneLM generates sequences without fixing the gaps. We do not anticipate these substantially affect the performance of CloneBO and can potentially be addressed in future work.
>
> The increased Fv mutation rate may be due to (1) the increased rate of Fv mutations in the mature BCRs that CloneLM is trained on, or (2) it may be reverting mutations from germline that are present in the conditional $X_0$; neither of these necessarily represent a pathology. For (1), CloneLM is trained on annotated clonal families from FastBCR, which should exclude naive BCRs which have not undergone somatic hypermutation. Olsen et al., 2024 argue that reflecting the mutational spectrum of mature BCRs is favourable for drug design. For (2), CloneLM may recognize mutations from germline in $X_0$ and generate a clone with many sequences that exclude that germline mutation; in this case, a single “reversion” may appear as a mutation in every sequence in the clone that is not $X_0$ in our Fig. 2A. Since mutations from germline can be passenger mutations or transient, it is not necessarily pathological that CloneLM may revert these mutations in some cases.
>
> When given a clone with gaps at the beginning, CloneLM may recognize that other sequences in the clone may also have come from a short-read sequencing machine and not fill in the gaps. On the other hand, we have noticed that when given a full sequence during optimization, CloneLM rarely introduces large sequencing errors or pathologies in its generation; since we also restrict sequences we test in lab to a few mutations form $X_0$, it is not obvious that CloneLM reproducing limited sequencing coverage present in the data should affect the performance of CloneBO.
>
> On the other hand, should we want to remove sequencing errors from the training data, we could inpute the gaps in coverage with models such as Olsen et al. 2022 in the training data; then we could train CloneLM on this data with imputed gaps.
>
> >The melting temperature and binding oracle experiment uses VHHs, which I would say is not ideal. While humanized VHHs do look kinda like human VH chain, they are still different, even in trivial ways (e.g. they often have much longer CDR H3). It's nice that the proposed method deals with this distribution shift, but it's not entierly fair to compare to the baselines, that were only developed for 'normal' Abs.
>
> Indeed, while ideally we would have validated on optimizing human VHs, we validated on data from experiments on humanized VHHs since this is the data we had available. We note however that both CloneLM and LaMBO-Ab are trained on “normal Abs” data from human OAS, so CloneBO does not have an unfair advantage from, for example, having seen Camelid VHHs.
> In the paper you write "de novo design method DiffAb", but DiffAb it's not de novo, it needs a co-crystal structure as a starting point.
>
> In our experiments we use DiffAb to iteratively optimize an antibody by suggesting mutations. Therefore, indeed, we don’t use it for de novo design and in the new draft we have removed 3 references to it as such.
>
> >Figure 4 (b) second plot, only 1 point beats best point of Lambo, the average of Lambo also looks better so I'm not fully convinced that the proposed method produces more stable antibodies. Of course, the KD results are quite convincing.
>
> Indeed, while we noticed the most dramatic improvement from the baseline in silico for stability rather than binding, the situation was reversed in vitro. One hypothesis for why this is the case is that our trained oracles didn’t represent the variance in the in vitro measurements. We tuned our noise parameter $\sigma$ to generate realistic clones conditioned on the starting pool of the in silico data (note: this procedure is sound as it does not use any test data); we noticed the tuned value of $\sigma$ was the same for both binding and stability starting pools and therefore reused this value for all experiments. We hypothesize that re-tuning this parameter on the real in vitro starting pool data may have resulted in better predictions.
>
> >When training the ClonalPLM are the clonal families (their order) re-mixed (e.g. for Light chain as it uses many epochs) if not, how is the ordering chosen?
>
> Yes, the sequences in the clone are shuffled before being fed to CloneLM.

---

> > ### Author Response · Authors · 2024-11-21
> > **Comment 2**
> >
> > >What is the edit distance distribution for the in-vitro designs? In the method description (Section 6.4) you state "iteratively optimize F(X) over the top substitution for up to 3 substitutions", does this mean that designs can only have maximum edit distance of 3 in your experiments? If yes were the baselines constrained to the same edit distance?
> >
> > Yes, our designs have a maximum edit distance of 3. We considered this largely as a technique to limit using large amounts of compute optimizing rather than a technique to regularize our designs. First, even without the explicit limit, our designs often did not end up more than 3 mutations away: of the 200 sequences designed for testing in vitro roughly one third had an edit distance of 1, one third an edit distance of 2 and one third and edit distance of 3; note the best performing binding and stability sequences both had an edit distance of 1. In a new experiment in the middle column of Fig. 14, we note that CloneBO roughly performs the same with an edit threshold of $L=1$, $3$, or $5$.
> >
> > The baselines all consider different strategies to regularize their designs to be near previous sequences, which we did not modify. LaMBO for example considers a penalty on moving too far in the latent space, and in practice only suggested sequences with edit distance 1 in vitro.
> >
> > Citations:
> >
> > Olsen, Tobias H., Iain H. Moal, and Charlotte M. Deane. 2024. “Addressing the Antibody Germline Bias and Its Effect on Language Models for Improved Antibody Design.” bioRxiv. https://doi.org/10.1101/2024.02.02.578678.
> >
> > Olsen, Tobias H., Iain H. Moal, and Charlotte M. Deane. 2022. “AbLang: An Antibody Language Model for Completing Antibody Sequences.” Bioinformatics Advances 2 (1): vbac046.

---

> ### Comment · Reviewer_mvea · 2024-11-26
>
> Thank you for your rebuttal. I will keep my score.

---

### Official Review · Reviewer_gX85 · 2024-11-01

**Soundness:** 4
**Presentation:** 3
**Contribution:** 3
**Rating:** 8
**Confidence:** 4

**Summary:**

The paper introduces a method, termed CloneBO, for optimizing antibody sequences for stability and binding. Their approach uses an LLM to model the distribution of a collection of clonal populations of antibodies, drawing inspiration from the evolutionary trajectories of antibodies in the body to predict beneficial mutations. The authors introduce several ideas to improve sampling quality from their LLM, including sampling from a martingale posterior and constructing a twisted MC scheme that biases towards observed antibodies. Their approach is tested both on in silico benchmarks and in vitro experiments and shows outstanding performance in the design tasks they’ve attempted.

**Strengths:**

* Context sharing and placement of the solution: The exposition is strong (on the antibody design problem), code is shared, well-organized and readable. The paper targets an important problem and is lucid about how to get to impact in that domain.
* Innovation: Multiple interesting and recent ideas have been introduced together for this problem domain (Simulating clonal evolution, Martingale posteriors, combining twisted Montecarlo and LLMs), and the results are convincing in silico and in vitro.
* Benchmarking: multiple relevant design approaches have been benchmarked, and the method performs well comparatively.
* In vitro validation: The authors conduct extensive validation, both in silico and in vitro. In vitro validation is rare among ML papers but absolutely crucial in evaluating whether the method has practical use. The authors have done ablation studies on the effects of their sampling scheme and show each individual component helps the performance of the optimizer.

**Weaknesses:**

* The experimental data are not shared. This is not disqualifying for an ML paper in my view, but should be noted as a weakness.
* The paper is well structured but somewhat hard to read, partly due to the many ideas that it tries to cover in the main text. I suggest moving some of the mathematical exposition into the SI and describe the intuition more concisely but clearly.

**Questions:**

* I’m not clear on the exchangeability claim, evolutionary processes have a clear arrow of time, with sequences ordered in a way that can be predicted, unless we are only looking at the final leaves of the tree, which in my understanding is not necessarily true for immune populations. I appreciate a clearer exposition here.
* I’m unsure why the authors have chosen to model the light chain and heavy-chains separately. A clear explanation would be helpful.
* Line 39: “up to thousands of previous iterations” this is somewhat misleading, not all iterations are the same as in many cases these are done in batches (“measurements” are always done in batches), and batch-iterations rarely exceed 10. In silico iterations can reach thousands but the description here is blurry and especially “measurements” is best reserved for real world queries rather than oracle queries.

---

> ### Author Response · Authors · 2024-11-21
> **Comment**
>
> Thank you for your thoughtful review and suggestions! We address each of your suggestions and questions in detail below. In addition to these points, our new draft has improved the aesthetic of many figures, but kept the displayed data the same.
>
> >The paper is well structured but somewhat hard to read, partly due to the many ideas that it tries to cover in the main text. I suggest moving some of the mathematical exposition into the SI and describe the intuition more concisely but clearly.
>
> The most challenging section is the introduction of the twisted sampling. In the new draft we’ve therefore moved two paragraphs of the mathematical exposition in section 6.3 to the appendix and added a more intuitive explanation.
>
> >I’m not clear on the exchangeability claim, evolutionary processes have a clear arrow of time, with sequences ordered in a way that can be predicted, unless we are only looking at the final leaves of the tree, which in my understanding is not necessarily true for immune populations. I appreciate a clearer exposition here.
>
> Indeed, unlike protein family sequences, clonal family sequences are indeed not necessarily leaves of the tree. While this doesn’t affect exchangeability, it has the potential to affect the claim that the learned distribution has probability proportional to fitness by introducing bias and phylogenetic correlation. However phylogenetic correlations and biases are also present in protein datasets where the claim that the learned probability is proportional to fitness has been the basis for accurate mutation effect prediction. We hope the approximation is similarly accurate in our case. The evolutionary model used for CloneBO can in principle be improved by accounting for bias; as well, the presence of sequences which aren’t leaves in principle allows one to build models that can learn the direction of evolution, an exciting direction for future work.
>
> Formally, for protein modeling, the logic is that sequences in a protein family are (1) exchangeable and therefore come iid from some distribution that we can learn by training a machine learning model; next (2) it’s reasoned that, since these proteins are leaves of an evolutionary tree, this distribution has probability proportional to fitness (for example, Weinstein et al., 2022). The sequences we see from a clonal family are also exchangeable, so they must come iid from some distribution we can learn with a model. However, unlike protein family sequences, clonal family sequences are indeed not necessarily leaves of the tree; this impacts the logic of (2), the interpretation of the learned model.
>
> If antibody sequences in clonal families evolved forever and had little phylogenetic correlations, then population genetics suggests that their distribution is exactly in proportion to fitness. In principle, the fact that sequences in clonal families may not be leaves is not necessarily a problem – one can for example consider them samples from the same MCMC chain. In practice, the presence of sequences that are not leaves likely increases phylogenetic correlation, and bias from not evolving long enough. However, this bias is present in large scale phylogenetic correlation in protein families as well (Weinstein et al., 2022). Therefore we motivated CloneBO by suggesting (2) may also be a good approximation for clonal families.
>
> The CloneBO evolutionary model can in principle be improved by accounting for the bias that comes from evolving for a short amount of time, and by using sequences which are not leaves of the evolutionary tree to learn the arrow of evolution. First, for each clone, it is often possible to infer the ancestral naive sequence $\tilde X$; sequences in a clonal family are unlikely to evolve long enough to diverge substantially from this sequence so a potentially more accurate model accounting for this bias may by $\log p(X|\mathrm{clone})=F(X)-\mathrm{distance}(X, \tilde X)$. As well, once one infers $\tilde X$, immunologists often use the number of mutations in the framework region as an estimate for how long a sequence has been evolving; this potentially allows a model to learn the direction of evolution within each clonal family – something that is impossible to do in the protein case where all sequences are leaves.

---

> > ### Author Response · Authors · 2024-11-21
> > **Comment 2**
> >
> > > I’m unsure why the authors have chosen to model the light chain and heavy-chains separately. A clear explanation would be helpful.
> >
> > Rather than build two models to separately generate heavy and light chains, we could have trained a model to (1) generate either heavy or light chains, or (2) generate full antibodies – paired heavy and light chains.
> >
> > There are models that take the approach (1), such as IgLM (Shuai et al., 2023). Since there is a large amount of both heavy and light data however, we didn’t expect that we would observe a large improvement by training a single model on both sets of data. We also had the resources to train two models in parallel that could each focus on learning the heavy and light chain datasets separately. We suspect IgLM took this approach for the convenience of having both modalities in one model a practitioner could download.
> >
> > For (2), human antibodies are made up of pairs of heavy and light chains so it is most natural to model pairs. However, due to the limits of current sequencing technology, it is much easier to get all the heavy or light chain sequences in a patient’s repertoire than identify which sequences pair with which. This is manifest in the availability of the data, where in OAS, only roughly 0.1% of heavy chain sequences have a known paired light chain sequence. Learning from this very limited data while transferring the knowledge gained on the much larger unpaired data requires careful engineering such as the fine tuning approach recently employed by Kenlay et al., 2024. Indeed, this is an interesting direction for future work, especially as the amount of paired data grows.
> >
> > >Line 39: “up to thousands of previous iterations” this is somewhat misleading, not all iterations are the same as in many cases these are done in batches (“measurements” are always done in batches), and batch-iterations rarely exceed 10. In silico iterations can reach thousands but the description here is blurry and especially “measurements” is best reserved for real world queries rather than oracle queries.
> >
> > Indeed, in most cases each iteration involves measuring a large batch and the number of iterations is not in the thousands. In the new draft we’ve changed the sentence to “To make these predictions, we can learn from up to thousands of measurements of sequences from many previous iterations”.
> >
> > Citations:
> >
> > Shuai, Richard W., Jeffrey A. Ruffolo, and Jeffrey J. Gray. 2023. “IgLM: Infilling Language Modeling for Antibody Sequence Design.” Cell Systems 14 (11): 979-989.e4.
> >
> > Kenlay, Henry, Frédéric A. Dreyer, Aleksandr Kovaltsuk, Dom Miketa, Douglas Pires, and Charlotte M. Deane. 2024. “Large Scale Paired Antibody Language Models.” arXiv [q-Bio.BM]. arXiv. http://arxiv.org/abs/2403.17889.
> >
> > Weinstein, Eli N., Alan N. Amin, Jonathan Frazer, and Debora S. Marks. 2022. “Non-Identifiability and the Blessings of Misspecification in Models of Molecular Fitness and Phylogeny.” Advances in Neural Information Processing Systems, December.

---

### Official Review · Reviewer_9Ldu · 2024-11-03

**Soundness:** 3
**Presentation:** 3
**Contribution:** 3
**Rating:** 8
**Confidence:** 2

**Summary:**

In this paper authors propose CloneBO - a procedure which aims at streamlining the antibody sequence optimisation. The approach relies on a language model which training is heavily inspired by the evolutionary mechanisms present in the immune system and allows for iterative guidance of generation by taking into the account previously measured samples. Authors validate their approach both in silico and in vitro demonstrating significant improvements over other methods.

**Strengths:**

This paper focuses on antibody sequence optimisation, an important problem of multi-objective nature that is encountered in every drug discovery project. Approaches that improve this drug pipeline stage have a potential of streamlining development of antibody-based therapies by shortening the development process and decreasing its costs.
The main original contribution of this paper lies in jointly modelling the clonal families (groups of evolutionarily connected antibody sequences that are computationally inferred from large scale sequencing data).
Authors validate their approach on several benchmarks and showcase potential of improving individual relevant traits of antibodies like binding affinity, humanness and thermal stability. Reported results significantly outperform state-of-the-art methods. Improvements are reported both in silico (through better predicted scores of trained oracles) and with in vitro experiments, which significantly strengthens the contribution.

**Weaknesses:**

Method requires initial, viable sequence i.e. the starting point from which the optimisation begins. This limits the applicability in some design scenarios where such sequence is not known and therefore hinders the impact of the approach compared to parallel lines of research that focus on e.g. structure based binder design and optimisation.

**Questions:**

Training of CloneLM relies on a data pre-processing step (grouping of sequences into clonal families) done with FastBCR. Since this is a critical initial step I wonder if the authors examined the effects of different preprocessing tools or hyperparameters on downstream performance or at least, given the large resource demands of LM training, the changes in distribution of processed data.

---

> ### Author Response · Authors · 2024-11-21
> **Comment**
>
> Thank you for your thoughtful and thorough review! We address your points and questions below. In addition to these points, we’ve made a number of aesthetic changes to the figures of the paper in the new draft.
>
> > Method requires initial, viable sequence i.e. the starting point from which the optimisation begins. This limits the applicability in some design scenarios where such sequence is not known and therefore hinders the impact of the approach compared to parallel lines of research that focus on e.g. structure based binder design and optimisation.
>
> Indeed CloneBO requires a starting point $X_0$ for optimization. This can be contrasted to other methods which for example take in a structure of a bound antibody and suggest sequences de novo that may adopt that structure.
>
> There have been several recent advances in de novo design, both computational or experimental. But it is almost always the case that these designs are not immediately suitable as drugs either because they have too little activity or are not stable in the human body; to build a drug, these designs must be optimized. We see de novo design methods, computational or experimental, as building $X_0$ and CloneBO as complementing these methods by optimizing their outputs.
>
> > Training of CloneLM relies on a data pre-processing step (grouping of sequences into clonal families) done with FastBCR. Since this is a critical initial step I wonder if the authors examined the effects of different preprocessing tools or hyperparameters on downstream performance or at least, given the large resource demands of LM training, the changes in distribution of processed data.
>
> Indeed CloneBO is trained on the outputs of FastBCR, which depend on the hyperparameters used for FastBCR. We used the default hyperparameters of FastBCR which were optimized for annotation accuracy on simulated data in their publication. We manually inspected select annotated clones to ensure they contained related sequences while different clones contained sufficiently different sequences. Given the computational cost of annotating all of OAS and training a model on this data, as well as the absence of obvious pathologies in the manually inspected clones, we decided to leave this exploration to future work.

---

### Official Review · Reviewer_Bdzt · 2024-11-04

**Soundness:** 2
**Presentation:** 2
**Contribution:** 3
**Rating:** 5
**Confidence:** 3

**Summary:**

This paper treats antibody design from pure sequence view, using clonal family to guide the model. The paper shows CloneBO can optimize antibody sequences better than former methods. Outstandingly, some experiments in vitro also support the effectiveness, which is often ignored in similar works.

**Strengths:**

1. The idea of applying martingale in antibody optimization is novel, and surprisingly fits the nature of evloving process of antibody.
2. Wet lab experiment is a highlight.

**Weaknesses:**

1. The experiment seems weak. Many influential works in recent years are not taken into comparison, such as[1][2]. They authors claim that structure based de novo design cannot make use of previous measurements and must have access to structure so they are not suitable for this task, which I highly suspect. Antibody is a specific protein type that highly reliable on its structure to perform. Even though stucture data are scarse, authors should demonstrate CloneBO’s supriority by showing that using massive sequence data can lead to higher performance only using sequence.
2. The representation is vague. Without any specific table, it’s hard to comprehend directly. This could also be a potential problem for following works to follow.

[1]Kong X, Huang W, Liu Y. End-to-end full-atom antibody design[J]. arXiv preprint arXiv:2302.00203, 2023.

[2]Lin H, Wu L, Yufei H, et al. GeoAB: Towards realistic antibody design and reliable affinity maturation. ICML2024.

**Questions:**

1. Does the optimization process in Fig. 3a converge? It seems that fitness is still rising in the end.
See weaknesses.

---

> ### Author Response · Authors · 2024-11-21
> **Comment**
>
> Thank you for your thoughtful review! We address each of your questions and suggestions below. We have also modified the figures of the text to make them more aesthetically pleasing without changing their data.
>
>
> >The experiment seems weak. Many influential works in recent years are not taken into comparison, such as[1][2]. They authors claim that structure based de novo design cannot make use of previous measurements and must have access to structure so they are not suitable for this task, which I highly suspect. Antibody is a specific protein type that highly reliable on its structure to perform. Even though stucture data are scarse, authors should demonstrate CloneBO’s supriority by showing that using massive sequence data can lead to higher performance only using sequence.
>
> Unfortunately the majority of structure design methods are not immediately applicable to all iterative design settings and even methods such as [1, 2] come with some conceptual downsides. Nevertheless we were also interested in the question of comparing them when they are applicable, so, as we describe below, in App C.4 we built a setting in which structure-based methods can potentially be used for iterative design, compared to [1], and did not see them outperform methods built for iterative design.
>
> When making a drug, there are a huge number of methods that are built to find “hits” – molecules that have a bit of activity. In almost every case, these “hits” have too little activity or stability to act as a drug, so they require optimization; techniques for optimization are distinct from those that are used in practice for finding hits and this is the drug design step we target with CloneBO. The vast majority of structure-based design methods target the former problem – they take in a structure of a bound antibody and return designed sequences – this is pointed out by the reviewer’s citations [1, 2]. Since these methods target a different problem, they are not immediately applicable to iterative design.
>
> [1, 2] adapt de novo structure design methods to suggest mutations to existing sequences. However, they still suffer some conceptual downsides compared to our baselines built for iterative design: (1) they require a co-crystal structure as a starting point, (2) they directly optimize for binding and have no obvious way to optimize for other properties such as expressibility or stability, (3) there is no obvious way for them to learn from previous measurements, (4) they may fail if a crystal structure is not a good representation of binding dynamics in the lab.
>
> Our target in Fig 3 and 4 did not have a co crystal structure so we could not compare to these methods. We therefore devised another setting to compare to structure-based methods in App C.4 – optimizing for SARS CoV binding. We took an oracle used by Jin et al., 2022 to validate their structure-based design method and sought to optimize binding for sequences with available co-crystal structures. We compared CloneBO and our baselines to DiffAb [1]; we could not compare to GeoAB [2] as they do not have available code for iterative design. We used a greedy optimization method to optimize sequences with mutations suggested by DiffAb. Note in 3 / 6 cases we did not have a starting structure to give DiffAb. We see DiffAb underperformed both CloneBO and the baseline LaMBO; this could be because (4) structure is a poor prior for what this oracle is measuring, or (3) CloneBO and LaMBO are able to find and exploit useful patterns in the previous measurements while the DiffAb optimization routine is not.
>
> >Does the optimization process in Fig. 3a converge? It seems that fitness is still rising in the end. See weaknesses.
>
> When optimizing an antibody, one usually begins with a fixed budget and is interested in maximizing the improvement they can get with that budget. We mimic this setting in our experiments in Fig 3 – we have a budget of 100 steps to optimize the antibody as best we can. If we were given unlimited steps, all methods in Fig 3a would converge to the best sequence as they would test every realistic antibody sequence.
>
> >The representation is vague. Without any specific table, it’s hard to comprehend directly. This could also be a potential problem for following works to follow.
>
> In our new draft we’ve included a table with the exact values of the results of the in silico experiments in App. C.3 for reference by future practitioners.
>
> Citation:
>
> Jin, Wengong, Jeremy Wohlwend, Regina Barzilay, and Tommi Jaakkola. 2021. “Iterative Refinement Graph Neural Network for Antibody Sequence-Structure Co-Design.” In International Conference of Learning Representations 2022.

---

> > ### Comment · Reviewer_Bdzt · 2024-11-23
> >
> > I have read the responses and raised my score. I strongly recommend the authors include tables for the other figures and put them in the main text.

---

### Meta-Review · Area_Chair_xUSn · 2024-12-21

**Metareview:**

This paper proposes Clone-informed Bayesian Optimization (CloneBO), a Bayesian optimization procedure for antibody design. The interesting innovation comes from the way they capture the prior antibody distribution: They train an LLM on clonal families to mimic how the immune system diversifies antibodies. Notably, evaluations are performed in both simulated and **in vitro** experiments, showcasing the strong empirical potential of the proposed framework.

**Additional Comments On Reviewer Discussion:**

NA

---

### Decision · Program_Chairs · 2025-01-22

Accept (Spotlight)